# Glutamine catabolism supports amino acid biosynthesis and suppresses the integrated stress response to promote photoreceptor survival

Moloy T Goswami[1†], Eric Weh[1†], Shubha Subramanya[1], Katherine M Weh[1], Hima Bindu Durumutla[1,2], Heather Hager[1], Nicholas Miller[1], Sraboni Chaudhury[1], Anthony Andren[3], Peter Sajjakulnukit[3], Li Zhang[3], Cagri Besirli[1], Costas A Lyssiotis[3,4,5], Thomas J Wubben[1]*

[1]Department of Ophthalmology and Visual Sciences, University of Michigan, Ann Arbor, United States; [2]Molecular and Developmental Biology Graduate Program, Cincinnati Children's Hospital Medical Center, Cincinnati, United States; [3]Department of Molecular & Integrative Physiology, University of Michigan, Ann Arbor, United States; [4]Department of Internal Medicine, Division of Gastroenterology and Hepatology, University of Michigan, Ann Arbor, United States; [5]Rogel Cancer Center, University of Michigan, Ann Arbor, United States

*For correspondence:
twubben@med.umich.edu

[†]These authors contributed equally to this work

## eLife Assessment

Goswami and colleagues used rod-specific Gls1 (the gene encoding glutaminase 1) knockout mice to investigate the role of GLS1 in photoreceptor health when GLS1 was deleted from developing or adult photoreceptor cells. This study is **fundamental** as it shows the critical role of glutamine catabolism in photoreceptor cell health using in vivo model systems. The evidence supporting the authors' claims is **compelling**. The studies add new insight into how specific metabolites support vision.

**Abstract** Photoreceptor loss results in vision loss in many blinding diseases, and metabolic dysfunction underlies photoreceptor degeneration. So, exploiting photoreceptor metabolism is an attractive strategy to prevent vision loss. Yet, the metabolic pathways that maintain photoreceptor health remain largely unknown. Here, we investigated the dependence of photoreceptors on glutamine (Gln) catabolism. Gln is converted to glutamate via glutaminase (GLS), so mice lacking GLS in rod photoreceptors were generated to inhibit Gln catabolism. Loss of GLS produced rapid rod photoreceptor degeneration. In vivo metabolomic methodologies and metabolic supplementation identified Gln catabolism as critical for glutamate and aspartate biosynthesis. Concordant with this amino acid deprivation, the integrated stress response (ISR) was activated with protein synthesis attenuation, and inhibiting the ISR delayed photoreceptor loss. Furthermore, supplementing asparagine, which is synthesized from aspartate, delayed photoreceptor degeneration. Hence, Gln catabolism is integral to photoreceptor health, and these data reveal a novel metabolic axis in these metabolically demanding neurons.

## Introduction

Photoreceptor (PR) death is the cause of vision loss in many retinal diseases. A paucity of effective therapies exist that prevent PR death, so there is an unmet need for therapeutics that improve or prolong PR survival. The retina has a significant energetic demand, driven in large part by PRs (*Pan et al., 2021*). The prodigious bioenergetic requirements of PRs are due to the need to conduct phototransduction and neurotransmission as well as manufacture the lipid- and protein-rich outer segment, which is shed daily and phagocytosed by the retinal pigment epithelium (RPE; *Ng et al., 2015*; *Young, 1967*). PRs have little reserve capacity to generate adenosine triphosphate (ATP) and as a result, are adversely affected by small changes in energy homeostasis. As such, metabolic dysfunction has been shown to underlie PR cell death (*Bowne et al., 2006*; *Du et al., 2015*; *Hartong et al., 2008*; *Kooragayala et al., 2015*; *Pan et al., 2021*), and exploiting PR metabolism to make these cells more robust to stress is an attractive neuroprotective strategy (*Pan et al., 2021*). Yet, beyond glucose, the metabolic pathways integral to PR health remain largely unknown. This is a critical knowledge gap as identification of these pathways is likely to reveal new strategies for therapeutic intervention (*Duncan et al., 2018*).

Glucose is central to PR metabolism as these cells utilize aerobic glycolysis, or the conversion of glucose to lactate despite the presence of oxygen, for the production of both energy and anabolic building blocks, similar to cancer cells (*Aït-Ali et al., 2015*; *Chinchore et al., 2017*; *Kanow et al., 2017*; *Petit et al., 2018*; *Swarup et al., 2019*). Previous studies have shown that genetic knockdown of enzymes key to aerobic glycolysis leads to PR dysfunction and death (*Chinchore et al., 2017*; *Petit et al., 2018*; *Rajala, 2020*; *Weh et al., 2020*; *Wubben et al., 2017*). However, like other metabolically demanding cells, recent work has demonstrated that PRs have the flexibility to utilize fuel sources beyond glucose to meet their metabolic needs (*Adler et al., 2014*; *Daniele et al., 2022*; *Du et al., 2013b*; *Grenell et al., 2019*; *Joyal et al., 2016*; *Xu et al., 2020*).

Glutamine (Gln) is the most abundant amino acid found in the body and circulating within the blood (*Yang et al., 2017*), and many rapidly dividing cells, including cancer cells, which utilize aerobic glycolysis, depend on Gln for their survival and proliferation (*Yang et al., 2017*). Gln can serve as a substrate for multiple pathways, providing a carbon and nitrogen source for biosynthesis, energetics, and cellular reactive oxygen species (ROS) homeostasis (*Adler et al., 2014*; *Xu et al., 2020*; *Yang et al., 2017*). These features potentially make Gln an ideal alternative fuel source for PRs, whose bioenergetic demand rivals that of cancer cells despite being terminally differentiated (*Ng et al., 2015*). To this end, over half of the Gln in the retina is found in the outer retina, which is primarily composed of PRs (*Du et al., 2013b*; *Macosko et al., 2015*; *Voigt et al., 2020*). Ex vivo experiments have shown that Gln can be a source of carbons for TCA cycle intermediates in the retina and contribute to amino acid biosynthesis (*Du et al., 2013b*; *Grenell et al., 2019*; *Tsantilas et al., 2021*; *Xu et al., 2020*). Additionally, Gln was demonstrated to support nicotinamide adenine dinucleotide phosphate (NADPH) generation in the absence of glucose in vitro in isolated PRs (*Adler et al., 2014*). Finally, in a genetically altered mouse model that disrupts glucose transport to the PRs, both Gln and its transporter were upregulated in the retina, implying that metabolism of this amino acid may be supporting PR survival when glucose is limiting (*Swarup et al., 2019*).

Glutaminolysis is the process by which Gln is metabolized into TCA cycle intermediates for critical biosynthetic precursors. This process is initiated by the catabolism of Gln to glutamate (Glu) via one of two glutaminase enzymes: kidney-type glutaminase (GLS) or liver-type glutaminase (GLS2; *Yang et al., 2017*). Single-cell RNA sequencing data (GSE63473 and GSE142449) has demonstrated that GLS is the predominant isoform in the retina and PRs (*Macosko et al., 2015*; *Voigt et al., 2020*). Additionally, it has been shown that glutaminase activity is at least twofold higher in mitochondria-rich PR inner segments (*Ross et al., 1987*). Since the inner segments of PRs are responsible for supporting the majority of energy production and metabolism, GLS activity is likely critical for metabolic functions in PRs.

Glutaminolysis via GLS is indispensable to the metabolism of many cancer cells (*Altman et al., 2016*; *Yang et al., 2017*), and a variety of data show the similarities between cancer cell metabolism and retinal metabolism, and specifically, metabolism of the PRs in the outer retina (*Chinchore et al., 2017*; *Du et al., 2013b*; *Ng et al., 2015*; *Rajala, 2020*). Therefore, we hypothesized that GLS-initiated Gln catabolism may also be essential to PR metabolism, function, and survival. Previous studies utilizing in vitro or ex vivo methods with whole retinas from wild-type mice or mice with genetic perturbations

not confined to PRs *Adler et al., 2014*; *Du et al., 2013b*; *Grenell et al., 2019*; *Xu et al., 2020* have provided a foundation for this governing hypothesis, but none have examined the role of Gln catabolism specifically in PRs in vivo. In this study, we generated a rod photoreceptor-specific knockout of GLS to comprehensively study the importance of GLS-driven Gln catabolism in PRs.

## Results

### Generation of a rod photoreceptor-specific, *Gls* knockout mouse

To confirm the single-cell RNA sequencing data (*Macosko et al., 2015*; *Voigt et al., 2020*), we conducted real-time reverse transcription PCR (qRT-PCR) with primers specific for either *Gls* or *Gls2* (*Supplementary file 1*). These data show that *Gls* expression is 14 times greater than that of *Gls2* in the mouse retina (*Figure 1—figure supplement 1A*), indicating that GLS is indeed the predominant isoform in murine retina. Furthermore, retinal sections stained for GLS using immunofluorescence showed that GLS is expressed throughout the retina with enrichment in the PR inner segments (*Figure 1—figure supplement 1B*). It has been shown that GLS activity is at least twofold higher in the PR inner segments, which are rich in mitochondria (*Ross et al., 1987*). Accordingly, GLS segregated to the mitochondrial-enriched fraction more than the cytosolic fraction in the retina (*Figure 1—figure supplement 1C*).

We generated a rod PR-specific, *Gls* conditional knockout mouse to determine the role of GLS activity in PR survival and function. Animals homozygous for a floxed *Gls* allele and expressing a Cre-recombinase under the control of the rhodopsin promoter ($Gls^{fl/fl};Rho^{Cre+}$, cKO) as well as animals expressing only the Cre-recombinase ($Gls^{wt/wt};Rho^{Cre+}$, WT) were generated. Total retinal lysate was collected from cKO and WT animals at postnatal day 14 (P14) and GLS expression was measured (*Figure 1A*). These data show that cKO animals have significantly less GLS expression compared to WT animals. Whole eyes were collected at P14 and stained for GLS using immunofluorescence. *Figure 1B* shows significant loss of GLS expression in PR inner segments of cKO animals compared to WT animals. The remaining GLS in the inner segment layer is found within cone PRs, as shown in *Figure 1B* (arrows), confirming the knockout is specific to rod PRs. A lack of compensatory upregulation of *Gls2* in the retina was confirmed via qRT-PCR and in PRs using immunofluorescence (*Figure 1—figure supplement 1D–E*).

### Loss of GLS causes rapid PR degeneration

These data clearly show that GLS is significantly downregulated in cKO animals. To determine if loss of GLS effects PR survival, optical coherence tomography (OCT) was used to measure the in vivo thickness of each retinal layer at various timepoints in cKO and WT animals (*Figure 1C*). At P14, cKO and WT animals are indistinguishable in total retinal and outer nuclear layer (ONL) thickness (*Figure 1D and E*). However, by P21 a significant loss in total retinal, ONL, and IS/OS thickness is observed (*Figure 1D-F*) and cKO animals continue to experience loss of retinal, ONL and IS/OS thickness out to P84. Considering mice open their eyes around P14, we assessed if light exposure was contributing to PR degeneration. cKO mice were reared in the dark, but no change in the rate of ONL degeneration was observed when comparing dark-reared mice to those reared in 12 hr light/12 hr dark cyclic lighting conditions (*Figure 1—figure supplement 2*). Histology at P14, P21, and P42 confirmed loss of PR cell bodies (*Figure 1G and H*). As has been seen in other models of PR degeneration, glial fibrillary acidic protein (GFAP), which is a marker of stress-induced Müller glial cell activation, was increased in the retina of the cKO mouse at P21 and P42 (*Figure 1—figure supplement 3*; *Grenell et al., 2019*).

Next, to assess the cell death pathways contributing to PR degeneration in the cKO mouse, retinal sections from P21 animals were stained for TUNEL (*Figure 1I*). At P21, there is a significant increase in TUNEL-positive outer retinal cells in cKO mice compared to WT mice (*Figure 1J*). qRT-PCR analysis of cell-death-related gene expression at P14 (*Figure 1K*) demonstrated an increase in genes involved in apoptosis, necroptosis, and ferroptosis. These data coincide with the thought that apoptosis is the predominant mechanism of PR death in many retinal diseases, considering *Casp8* showed the greatest increase in expression in the cKO retina, but also that other cell death mechanisms can contribute to PR cell death (*Wubben et al., 2016*).

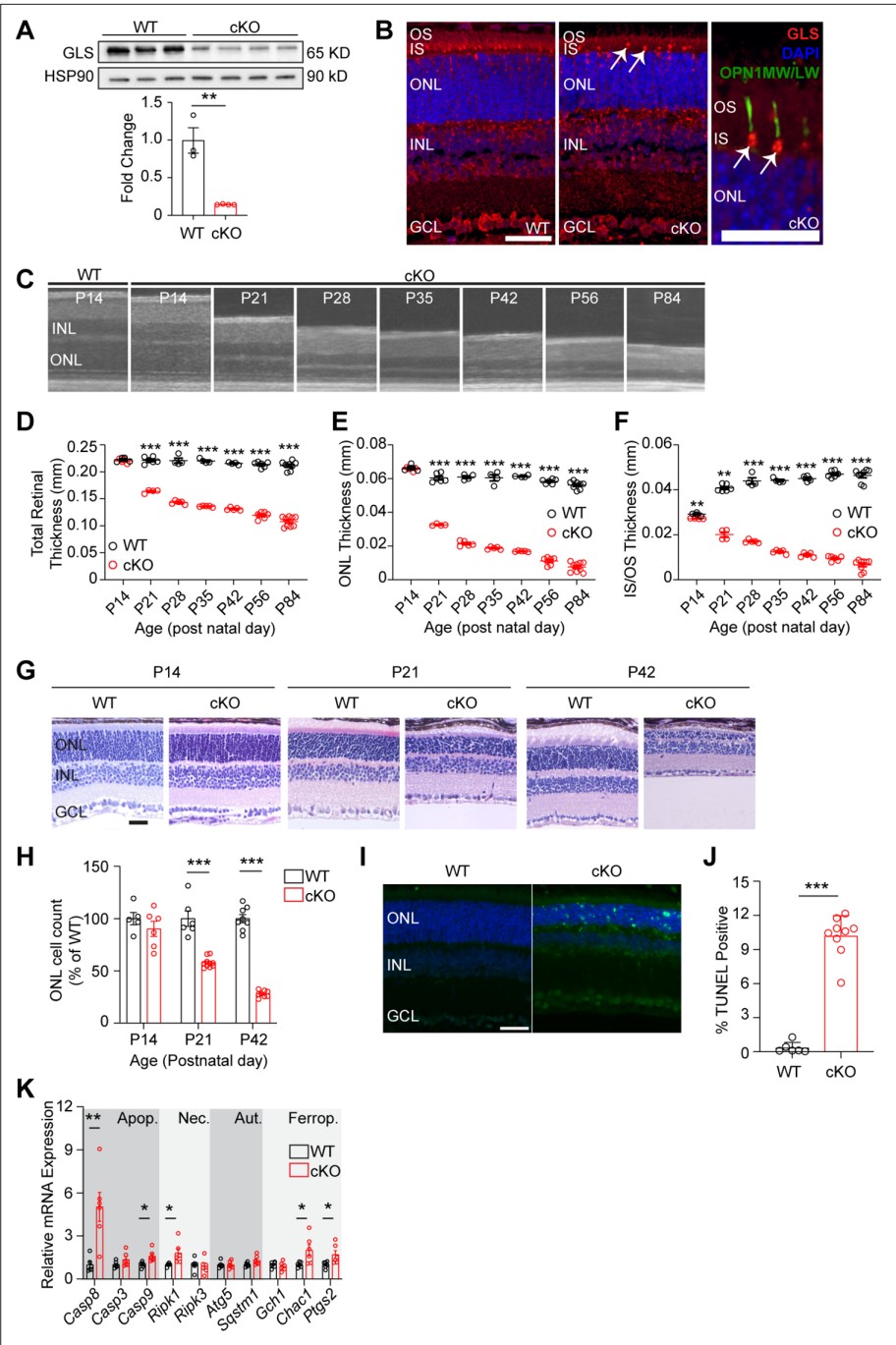

**Figure 1.** Rod photoreceptor-specific knockout of glutaminase (GLS) displays rapid retinal degeneration and increased markers of cell death. (**A**) Western blot analysis showing decreased GLS protein levels in the retina of WT and cKO mice at post-natal day 14 (**P14**). Quantitation of western blot results for N=3–4 animals per group. (**B**) Representative images for immunofluorescence of P14 mouse retinas (N=3 animals per group) stained for GLS (red), cone opsin (green), and nuclei (DAPI, blue) in WT and cKO mice. White arrows indicate remaining GLS expression in cone photoreceptors. Left and middle panel scale bars are 40 µm. Right image scale bar is 20 µm. (**C**) OCT images detailing outer retinal changes in cKO mice over time compared to WT. Retinal structures are comparable to WT mice at P14 but rapidly thin with age. (**D**) Total retinal thickness, (**E**) ONL thickness and (**F**) IS/OS thickness as determined by OCT in WT and cKO mice over time. N=4–9 eyes per group. (**G**) Representative hematoxylin and eosin stained retinal sections from rod photoreceptor-specific *Gls* conditional knockout (cKO) mice compared to wild-type (WT) mice at P14, P21, and P42. ONL, outer nuclear layer; INL, inner nuclear layer; GCL, ganglion cell layer. Scale bar is 40 µm. (**H**) ONL cell counts as a percent of WT retinas at P14, P21, and P42.

*Figure 1 continued on next page*

*Figure 1 continued*

N=5–10 eyes per group. (**I**) Representative images of WT and cKO retinas stained to detect TUNEL-positive cells (green) at P21. Scale bar is 40 µm. N=3–5 animals per group. (**J**) Quantitation of percent TUNEL-positive cells at P21 showing an increase in TUNEL-positive cells in cKO animals. N=3–5 animals per group. (**K**) qRT-PCR of genes related to cell death pathways including apoptosis (Apop.), necroptosis (Nec.), autophagy (Aut.), and ferroptosis (Ferrop) in WT and cKO mice at P14. N=6 animals per group. Statistical differences in (**A**), (**D**), (**E**), (**F**), (**H**), (**J**) and (**K**) are based on an unpaired two-tailed Student's t-test where *p<0.05, **p<0.01 and ***p<0.001. Data are presented as mean ± standard error of the mean. OCT: optical coherence tomography, OS: outer segment, IS: inner segment, ONL: outer nuclear layer, INL: inner nuclear layer, GCL: ganglion cell layer, TUNEL: terminal deoxynucleotidyl transferase dUTP nick and labeling.

The online version of this article includes the following source data and figure supplement(s) for figure 1:

**Source data 1.** Original western blot membranes corresponding to *Figure 1A*.

**Source data 2.** Unannotated western blot membranes corresponding to *Figure 1A*.

**Figure supplement 1.** GLS is the predominant isoform in the mouse retina and the rod photoreceptor-specific *Gls* conditional knockout mouse does not demonstrate compensatory upregulation of the *Gls2* isoform.

**Figure supplement 1—source data 1.** Original western blot membranes corresponding to *Figure 1—figure supplement 1C*.

**Figure supplement 1—source data 2.** Unannotated western blot membranes corresponding to *Figure 1—figure supplement 1C*.

**Figure supplement 2.** *Gls* conditional knockout mouse demonstrates similar degeneration under cyclic- and dark-reared conditions.

**Figure supplement 3.** Histology confirms increased Müller glial cell activation in rod photoreceptor-specific *Gls* conditional knockout mice.

**Figure supplement 4.** Rod photoreceptor-specific *Gls* conditional knockout mice have shorter outer segments.

**Figure supplement 5.** Rod photoreceptor-specific *Gls* knockout does not alter synaptic connectivity between photoreceptors and second-order neurons.

Retinas were stained for RHO expression using immunofluorescence, which indicated rod OSs were shorter after *Gls* knockout (*Figure 1—figure supplement 4A*). Because cKO mice demonstrated shorter OSs, transmission electron microscopy (TEM) was used to investigate the ultrastructure of cKO rod OSs (*Figure 1—figure supplement 4B–C*). The rod OSs in the cKO mouse appeared largely normal with well-organized, stacked disc membranes that maintained interdigitation with the RPE similar to the WT mouse, just shorter than WT rod OSs.

A deficiency in GLS activity could lead to a significant decrease in Glu, the main neurotransmitter used by PRs. TEM was utilized to assess the ultrastructure of rod PR ribbon synapses, which appear structurally intact in the remaining rod PRs of the cKO retina at P21 (*Figure 1—figure supplement 4D*). To further evaluate the synaptic connectivity between PRs and the inner retina, retinal sections from cKO and WT mice at P14 were labeled with wheat germ agglutinin (WGA), a plant lectin that binds N-acetylglucosamine and sialic acid residues, to highlight PR synaptic membranes and non-synaptic membranes (*McLaughlin et al., 1980*). Retinal sections from P14 WT and cKO mice stained with WGA did not demonstrate significant differences in the labeling of PR synaptic membranes in the outer plexiform layer (OPL; *Figure 1—figure supplement 5A*). Staining of P14 retinal sections from WT and cKO mice with an antibody against Bassoon, which labels the ribbon synapses of rods and cones in the OPL (*Kutsyr et al., 2021*), did not demonstrate differences in the synaptic connectivity between PRs and second-order neurons (*Figure 1—figure supplement 5B*). Additionally, retinal sections at P14, P21, and P42 were stained with antibodies against major cell-types in the inner retina, such as amacrine, ganglion, and bipolar cells, and no significant alterations in immunofluorescent patterns to suggest an inner retinal developmental abnormality were observed between cKO and WT animals (*Figure 1—figure supplement 5C*). Accordingly, inner retinal thinning was not observed until P42 (*Figure 1—figure supplement 5D*), when approximately 25% of cells remained in the ONL (*Figure 1H*). This thinning of the inner retinal area is likely secondary to the primary PR degeneration similar to that seen in other mouse models of retinal degeneration (*Ueta et al., 2012*).

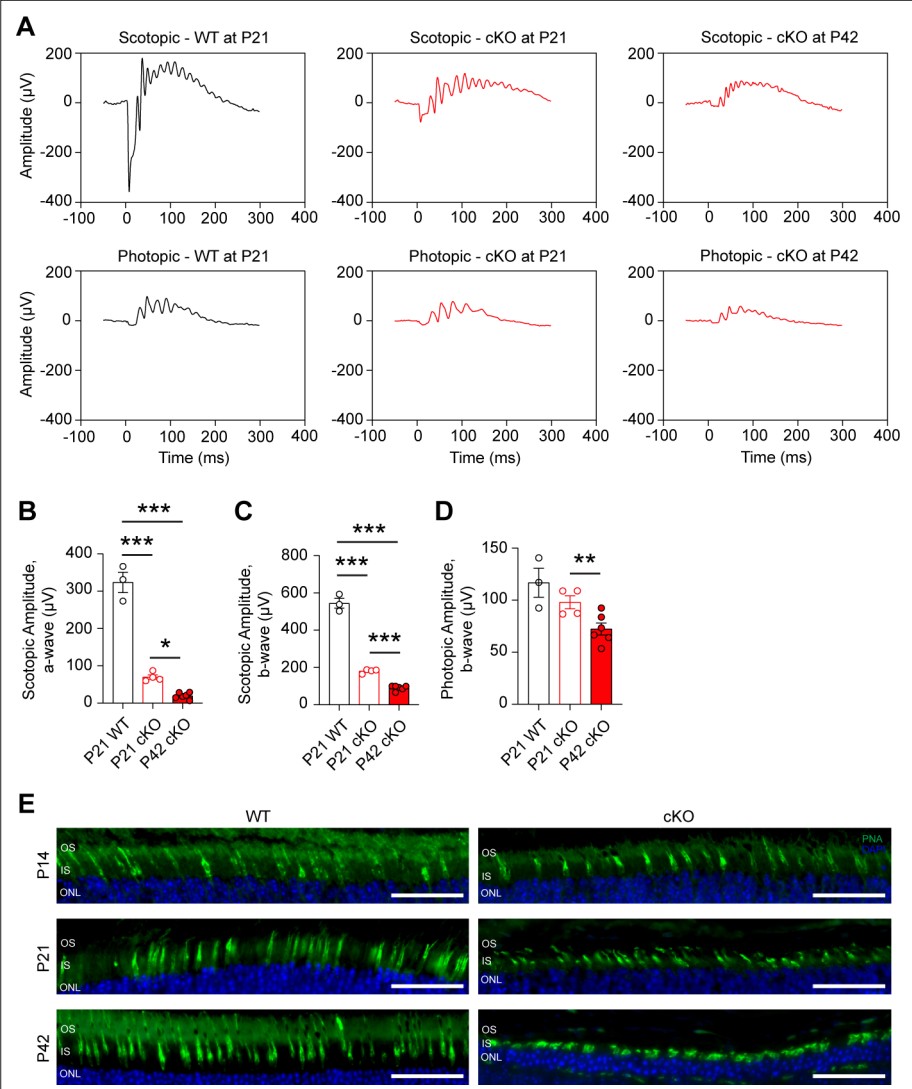

**Figure 2.** Loss of *Gls* in rod photoreceptors impairs retinal function. (**A**) Representative scotopic and photopic electroretinography (ERG) tracings for the rod photoreceptor-specific *Gls* conditional knockout mice (cKO) compared wild-type (WT) mice at P21 and P42. (**B**) ERG scotopic a-wave and (**C**) b-wave amplitudes in cKO mice compared to WT mice at P21 and P42. A flash intensity of 32 cd*s/m² was utilized. N=3–6 animals per group. (**D**) ERG photopic b-wave amplitudes in cKO mice compared to WT mice at P21 and P42. A flash intensity of 100 cd*s/m² was used. N=3–6 animals per group. (**E**) Representative images from staining of the cone-specific marker peanut agglutinin (PNA, green) and nuclei (DAPI, blue) in retinal sections from cKO mice compared to WT mice at P14, P21, and P42. Scale bars are 40 μm. Statistical differences in (**B**), (**C**) and (**D**) are based on an unpaired two-tailed Student's t-test where *p<0.05, **p<0.01 and ***p<0.001. Data are presented as mean ± SEM. ONL, outer nuclear layer; IS, inner segments; OS, outer segments. N=3 per group.

## Loss of *Gls* in rod photoreceptors impairs retinal function

Loss of IS/OS thickness, shorter OSs, and loss of PR cell bodies can result in functional loss. Thus, electroretinography (ERG) analysis was performed on cKO and WT animals at P21 and P42 (*Figure 2*). As expected, cKO animals show a significant loss in rod-driven scotopic a- and b-wave amplitudes at P21, which are further decreased by P42 (*Figure 2A-C*). Interestingly, a significant loss in photopic b-wave amplitude was also found at P42, suggesting a cone PR defect (*Figure 2D*). Immunofluorescent staining found that cone outer segments shorten as rod degeneration progresses from P14 to P42 (*Figure 2E*) supporting secondary cone degeneration, which is a common phenotype associated with rod-mediated retinal degeneration (*Caruso et al., 2020*).

## GLS is necessary for maintenance of mature photoreceptors

In mice expressing a Cre-recombinase under the control of the rhodopsin promoter, Cre-mediated excision of floxed genomic DNA has been observed as early as P7 (*Le et al., 2006*). To ensure the function of GLS is not restricted to this early phase of PR development and maturation and that it is critical to fully developed PRs as well, mice homozygous for a floxed *Gls* allele (*Gls^{fl/fl}*) and expressing an inducible Cre-recombinase under the control of the *Pde6g* promoter (*Gls^{fl/fl};Pde6g^{Cre:ERT2}*; *Koch et al., 2015*) as well as animals expressing only the inducible Cre-recombinase (*Gls^{wt/wt};Pde6g^{Cre:ERT2}*) were generated. This allowed for induction of Cre-recombinase activity in fully-mature PRs, specifically, by the administration of tamoxifen (TAM). Animals heterozygous for *Pde6g^{Cre:ERT2}* and homozygous for either the WT or floxed *Gls* allele (IND-cKO) were generated, and TAM was administered intraperitoneally for 5 consecutive days starting at P22 (*Figure 3*) as retinal development is typically considered complete by P21 (*Zhou et al., 2021*). Significant reduction of GLS in the retina of IND-cKO mice after TAM induction was confirmed via western blot and in PRs with immunofluorescence analysis (*Figure 3A and B*). Ten days after TAM induction, IND- cKO mice began to show a thinning of the IS/OS layer that preceded and then paralleled the rapid ONL degeneration observed on longitudinal OCT and histology (*Figure 3C-G*). Hence, GLS is also critical for the survival of fully developed PRs.

Beyond survival, GLS is also critical for PR function. ERG analyses performed 10 days after TAM induction, a time prior to major structural changes in the outer retina, demonstrated statistically significant reductions in the IND-cKO scotopic a- and b-waves as compared to the WT (*Figure 3H*). Similarly, photopic ERG demonstrated statistically significant decreases in the b-wave of the IND-cKO retina (*Figure 3I*). These data suggest that GLS-driven Gln catabolism plays a significant role not only in rod PR survival but their function as well. Additionally, *Pde6g* is expressed by rods to a significant degree but also by cones (*Voigt et al., 2020*). Therefore, the IND-cKO likely knocks out GLS from both rods and cones, which is in accordance with the immunofluorescence image in *Figure 3B* where GLS is not observed in rod or cone inner segments unlike in *Figure 1B* where GLS remains in cones. Hence, the reduction in IND-cKO photopic b-wave may suggest that GLS-driven Gln catabolism in cones impairs their synaptic transmission.

## GLS knockout does not alter nucleotide metabolism

Gln, and its breakdown product, Glu, have several fates in cellular metabolism (*Figure 4A*), which may underly the rapid and significant PR degeneration observed when GLS is knocked out of rod PRs. PRs require nucleotides to support phototransduction as well as transcriptional efforts for the continued synthesis of OSs. The catabolism of Gln to Glu by GLS produces an ammonium ion that can be used for the synthesis of nucleotides (*Yoo et al., 2020*). Additionally, inhibition of GLS has previously been demonstrated to inhibit pyrimidine and purine biosynthesis (*Alkan et al., 2018*; *Okazaki et al., 2017*). To explore the role of GLS-initiated Gln catabolism in nucleotide metabolism, in vivo targeted liquid chromatography-tandem mass spectrometry (LC-MS/MS) metabolomics were performed on P14 retina from WT and cKO animals (*Supplementary file 2*). No differences in key purine or pyrimidine metabolism intermediates, such as ribose 5-phosphate (R5P), inosine monophosphate (IMP), and uridine monophosphate (UMP), were observed between the WT and cKO retina (*Figure 4B*), suggesting that GLS-mediated catabolism of Gln is not a critical pathway for nucleotide synthesis in rod PRs.

## Redox balance is altered upon knockout of GLS in rod photoreceptors

Gln can also be used for the generation of the antioxidant molecule glutathione as well as in a non-canonical NADPH generating pathway (*Son et al., 2013*; *Yoo et al., 2020*). Glutathione is a tripeptide of glutamate, glycine, and cysteine. Glutamate is a product of the GLS reaction, and previous studies have demonstrated that inhibiting GLS reduces glutathione levels and increases ROS (*Li et al., 2015*). As such, the NADP$^+$/NADPH ratio was statistically significantly increased by 16% in the P14 cKO retina as compared to WT (*Figure 4C*), and the relative abundance of oxidized glutathione (GSSG) was decreased in the P14 cKO retina as compared to WT (*Figure 4D*). While an increased NADP$^+$/NADPH ratio in the cKO retina may imply that the abundance of GSSG should be increased in the cKO retina since reduced glutathione can be recovered from the oxidized form by the conversion of NADPH to NADP$^+$, the levels of GSSG have been seen to be reduced in other models where GLS is pharmacologically inhibited or genetically knocked down (*Daemen et al., 2018*). The lower levels of GSSG

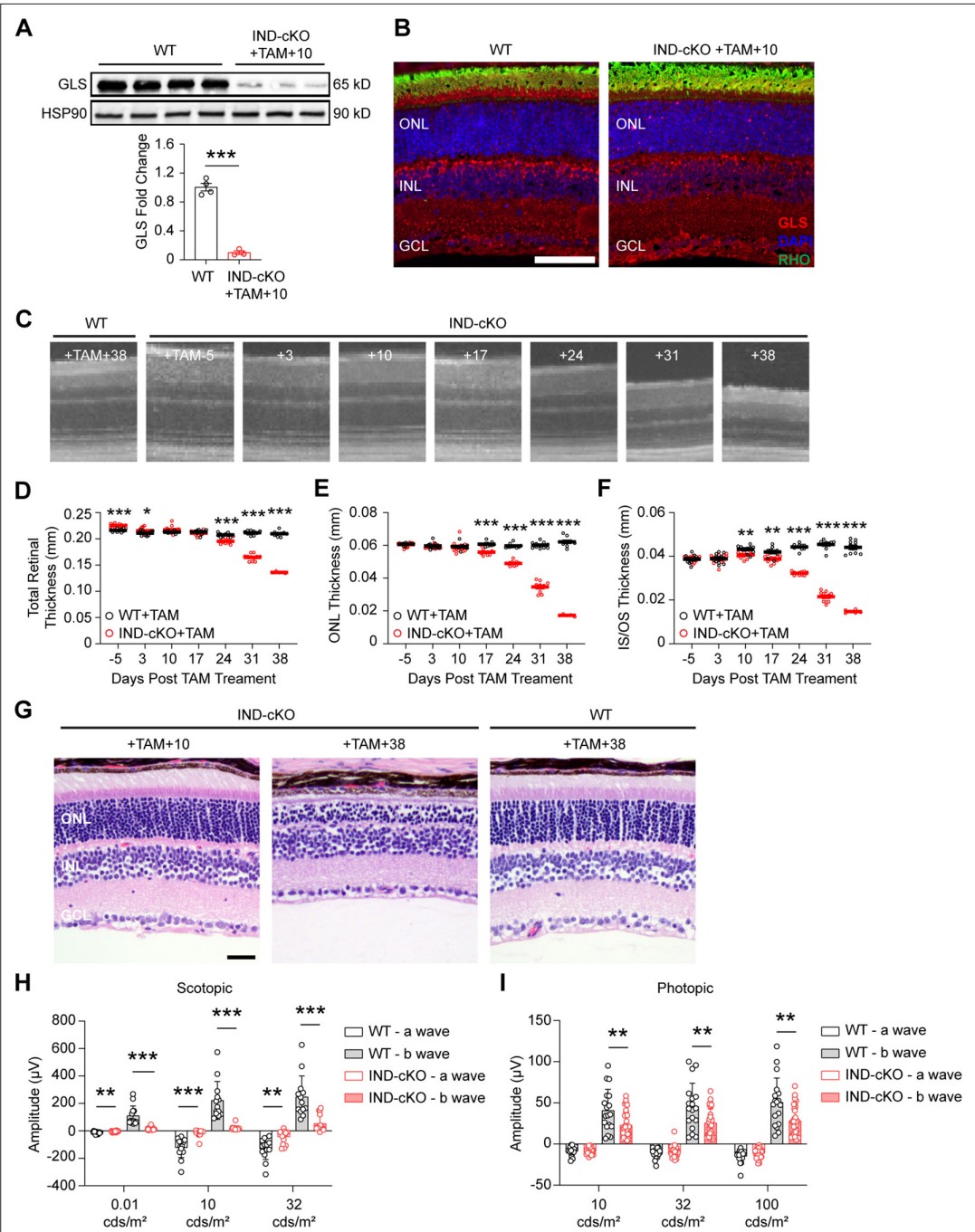

**Figure 3.** Rod photoreceptors require GLS for maintenance and maturation. *Gls*^fl/fl^ mice carrying a tamoxifen (TAM)-inducible Cre-recombinase (IND-cKO) under the control of *Pde6g* (*Gls*^fl/fl^;*Pde6g*^Cre:ERT2^) compared to mice expressing only the inducible Cre-recombinase (*Gls*^wt/wt^;*Pde6g*^Cre:ERT2^; WT). Both IND-cKO and WT were administered tamoxifen for 5 days starting at P22. (**A**) Quantitation of western blot results showing decreased GLS protein levels in the retina of IND-cKO animals at 10 days after tamoxifen induction compared to WT mice. HSP90 was used as a loading control. N=3–4 animals per group. (**B**) Representative GLS immunofluorescence (red) in IND-cKO mice compared to the wild-type (WT) mouse 10 days after tamoxifen induction. N=3 animals per group. Scale bar is 40 μm. (**C**) OCT images detailing outer retinal changes in WT and IND-cKO animals over time. Total retinal thickness (**D**), outer nuclear layer (ONL) thickness (**E**) and inner segment/outer segment (IS/OS) thickness (**F**) as determined by OCT for 38 days post tamoxifen. N=5–10 eyes per group. (**G**) Representative hematoxylin and eosin-stained retinal sections from IND-cKO mice compared to WT mice at 10 and 38 days after tamoxifen induction. N=3 animals per group. Scale bar is 40 μm. ONL, outer nuclear layer; INL, inner nuclear layer; GCL,

*Figure 3 continued on next page*

*Figure 3 continued*

ganglion cell layer. Quantitation of scotopic (**H**) and photopic (**I**) ERG a- and b-waves in WT and IND-cKO retina 10 days post TAM. N>12 eyes per genotype. Statistical differences in (**A**), (**D**), (**E**), (**F**), (**H**), and (**I**) are based on an unpaired two-tailed Student's t-test where *p<0.05, **p<0.01 and ***p<0.001. Data are presented as mean ± SEM.

The online version of this article includes the following source data for figure 3:

**Source data 1.** Original western blot membranes corresponding to *Figure 3A*.

**Source data 2.** Unannotated western blot membranes corresponding to *Figure 3A*.

---

may signal an overall reduction in the biosynthesis of glutathione considering the product of the GLS reaction, glutamate, is directly necessary for its synthesis as well as indirectly responsible for cysteine via the SLC7A11 (xCT) cystine/glutamate antiporter (*Yoo et al., 2020*). The expression of *Slc7a11* was reduced in the P14 cKO retina (*Figure 4E*). Hence, the biosynthesis of glutathione in rod PRs may be affected directly and indirectly when GLS is knocked out. In accordance with the antioxidants NADPH and glutathione potentially being reduced, the expression of the antioxidant enzymes *Sod1* and *Sod2*, which detoxify superoxide radicals, were increased in the P14 cKO retina (*Figure 4E*). These data suggest that GLS-mediated Gln catabolism regulates redox balance in rod PRs, and the altered redox balance that is a consequence of its knockout may be one factor contributing to PR degeneration in the GLS cKO mouse.

## Relative abundance of TCA cycle intermediates and mitochondrial function mostly unchanged in *Gls* cKO retina

While the changes in certain redox elements were statistically significant, they were modest alterations and may not account in full for the significant and rapid PR degeneration noted in the GLS cKO mouse. Many cancer cells that utilize aerobic glycolysis rely on Gln to replenish TCA cycle intermediates, which maintain oxidative metabolism and provide biosynthetic precursors. Considering PRs utilize aerobic glycolysis, similar to cancer cells, and have significant bioenergetic demands, we next sought to determine if rod PRs also depend on GLS-initiated glutaminolysis for TCA cycle metabolites and oxidative metabolism. Interestingly, LC-MS/MS-based targeted metabolomics demonstrated very few changes in the relative pool sizes of TCA cycle metabolites in the cKO compared to WT retina at P14 with only malate showing a statistically significant decrease (*Figure 4F*). Since GLS is enriched in the PR mitochondria (*Figure 1—figure supplement 1B*) and previous studies demonstrated that inhibiting GLS in certain cell lines reduces mitochondrial function (*Alkan et al., 2018*), mitochondrial stress tests were also performed on P14 cKO and WT retina, prior to PR loss, using the BaroFuse (*Kamat et al., 2023*). The basal oxygen consumption rate (OCR) as well as the changes in OCR in response to oligomycin or carbonyl cyanide 4-(trifluoromethoxy)phenylhydrazone (FCCP) were not statistically significantly different between WT and cKO retina (*Figure 4G and H*). The expression of complexes involved in oxidative phosphorylation were also unchanged between the WT and cKO retina at P14 (*Figure 4I*).

As stated earlier, a Cre-recombinase under the control of the rhodopsin has been shown to be activated as early as P7 (*Li et al., 2005*). To circumvent this early knockout of GLS, rod PRs may rewire their metabolism, as has been seen in other conditional knockout mouse models with this Cre-recombinase system (*Subramanya et al., 2023*; *Wubben et al., 2017*), to utilize different fuel sources for the maintenance of the TCA cycle and mitochondrial function. To this end, the expression of genes involved in glycolysis, pyruvate metabolism, and the TCA cycle were examined in the P14 cKO and WT retina before PR degeneration. Numerous genes were significantly altered in the cKO retina across the metabolic pathways (*Figure 4—figure supplement 1*). The expression of multiple genes in glycolysis and pyruvate metabolism were increased (*Figure 4—figure supplement 1A*), possibly suggesting that rod PRs are stimulating glucose oxidation to maintain the TCA cycle and mitochondrial function similar to that seen in certain cancer cells when GLS is inhibited (*Okazaki et al., 2017*). In contrast, the only significant change in expression of these genes in eyecup tissue was *Pdk4,* with a significant decrease in cKO vs WT mice (*Figure 4—figure supplement 2*). However, targeted metabolomics demonstrated only a minor change in glycolytic intermediates between the cKO and WT retina at P14 (*Figure 4F*). As targeted metabolomics at the P14 timepoint provides only a snapshot of the pool size, additional analyses with stable isotope tracing metabolomics are necessary

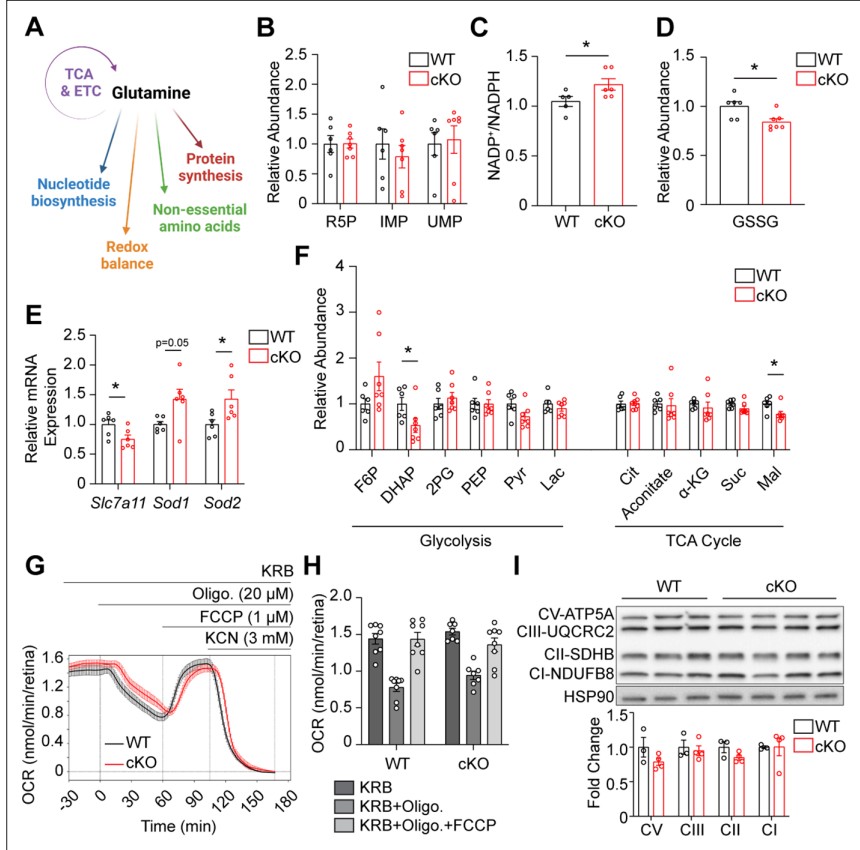

**Figure 4.** GLS cKO mice maintain levels of nucleotide, glycolytic and TCA cycle metabolites and mitochondrial function but demonstrate altered redox balance. (**A**) Schematic summarizing the biosynthetic and bioenergetic roles of glutamine. (**B**) Relative abundance of key intermediates in nucleotide metabolism in the retina of WT and cKO mice at P14 as determined by targeted metabolomics. N=6–7 animals per group. Relative abundance is the ion intensity normalized to the WT. (**C**) The NADP$^+$/NADPH ratio, as determined by bioluminescence assay, is significantly increased in the cKO as compared to WT retina at P14. N=5–6 animals per group. (**D**) Relative abundance of GSSG in the retina of WT and cKO mice at P14, prior to PR degeneration, as determined by targeted metabolomics. N=6–7 animals per group. (**E**) qRT-PCR of genes related to redox homeostasis are significantly altered in cKO compared to WT mice. N=6 animals per group. (**F**) Relative abundance of metabolites in glycolysis and the TCA cycle in WT and cKO retina at P14. N=6–7 animals per group. (**G**) Mitochondrial stress test carried out on isolated WT and cKO retina at P14 using the BaroFuse. The baseline was established by perifusing the tissue for 90 min and then oligomycin, FCCP, and KCN were injected into the perifusate sequentially as indicated. (**H**) Comparison of the effects of oligomycin and FCCP on OCR in P14 WT and cKO retina. N=6–8 animals per group. (**I**) Western blot analysis and quantitation of the mitochondrial electron transport chain complexes show no differences between WT and cKO retina. N=3–4 animals per group. Fold change is in relation to WT. Statistical differences in (**B–F**), (**H**) and (**I**) are based on an unpaired two-tailed Student's t-test where *p<0.05. Data are presented as mean ± SEM. R5P: ribose 5-phosphate, IMP: inosine monophosphate, UMP: uridine monophosphate, GSSG: glutathione disulfide, F6P: fructose 6-phosphate, DHAP: dihydroxyacetone phosphate, 2 PG: 2-phospho-D-glycerate, PEP: phosphoenolpyruvate, Pyr: pyruvate, Lac: lactate, Cit: citrate, α-KG: alpha-ketoglutarate, Suc: succinate, Mal: malate, Oligo: oligomycin, FCCP: carbonyl cyanide p-trifluorom ethoxyphenylhydrazone, KCN: potassium cyanide, CI-NDUFB8: complex 1, NADH:ubiquinone oxidoreductase subunit B8, CII-SDHB: complex 2, succinate dehydrogenase complex iron sulfur subunit B, CIII-UQCRC2: complex 3, ubiquinol-cytochrome c reductase core protein 2, CV-ATP5A: complex 5, ATP synthase F1 subunit alpha, HSP90: heat shock protein 90.

The online version of this article includes the following source data and figure supplement(s) for figure 4:

**Source data 1.** Original western blot membranes corresponding to *Figure 4I*.

**Source data 2.** Unannotated western blot membranes corresponding to *Figure 4I*.

*Figure 4 continued on next page*

*Figure 4 continued*

**Figure supplement 1.** Loss of *Gls* in rod photoreceptors alters the expression of retinal metabolism-related genes.

**Figure supplement 2.** Loss of *Gls* in rod photoreceptors does not alter the expression of metabolism-related genes in eyecups.

to assess nutrient utilization, in this case glucose. Uniformly labeled, $^{13}C_6$-glucose was intraperitoneally injected in P14 WT and cKO mice, and the retina harvested 45 min later for metabolomic analysis via LC-MS/MS (*Yuan et al., 2019*). No change in the fractional labeling of glycolytic intermediates was observed between the WT and cKO retina, and more so, an increase in the fractional labeling of TCA

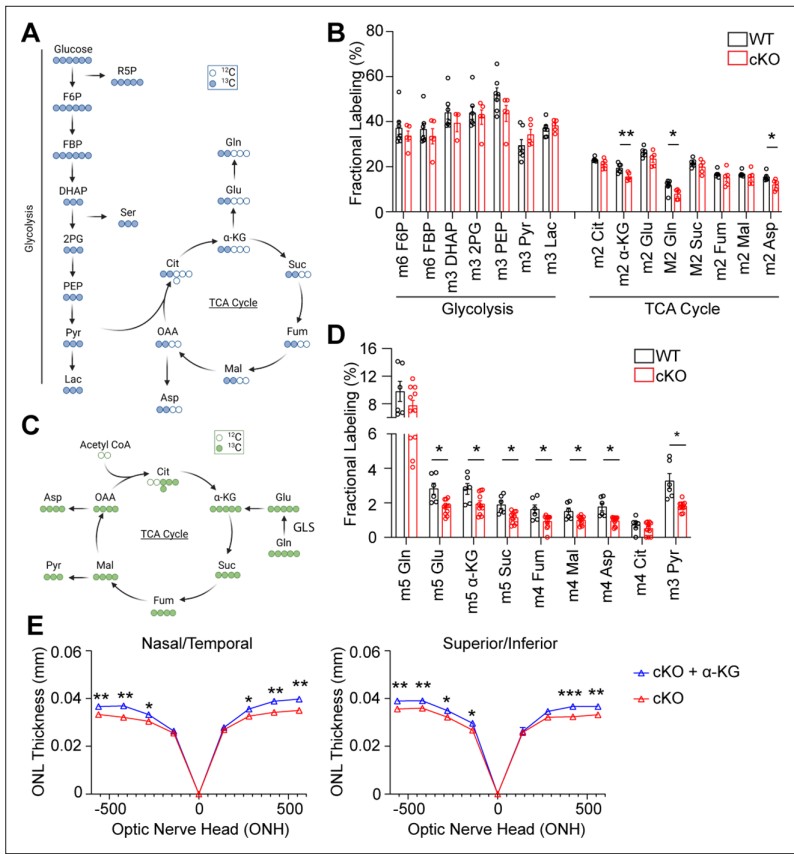

**Figure 5.** Loss of GLS in rod photoreceptors has significant effects on TCA cycle metabolism with only partial rescue upon α-KG supplementation. (**A**) Schematic summarizing $^{13}C_6$-glucose labeling in glycolytic and TCA cycle intermediates. (**B**) Fractional labeling of glycolytic and TCA cycle metabolites in the retina following intraperitoneal injection of $^{13}C_6$-glucose in WT and cKO mice at P14. N=5–6 animals per group. (**C**) Schematic summarizing $^{13}C_5$-Gln labeling in the TCA cycle. (**D**) Fractional labeling of TCA cycle metabolites in the retina following intraperitoneal injection of $^{13}C_5$-Gln in WT and cKO mice at P14. N=6–11 animals per group. (**E**) ONL thickness in cKO mice at P22 as assessed by OCT following α-KG supplementation (10 mg/mL) or vehicle (water) in the drinking water from P4-P22. N=5 animals per group. Statistical differences in (**B**), (**D**) and (**E**) are based on an unpaired two-tailed Student's t-test where *p<0.05, **p<0.01 and ***p<0.001. Data are presented as mean ± standard error of the mean. F6P: fructose 6-phosphate, FBP: fructose 1,6-bisphosphate, DHAP: dihydroxyacetone phosphate, 2 PG: 2-phosphoglycolate, PEP: phosphoenolpyruvate, Pyr: pyruvate, Lac: lactate, Gln: glutamine, Glu: glutamate, α-KG: alpha-ketoglutarate, Suc: succinate, Fum: fumarate, Mal: malate, OAA: oxaloacetate, Cit: citrate, Asp: aspartate.

The online version of this article includes the following figure supplement(s) for figure 5:

**Figure supplement 1.** Mass isotopologue distribution of $^{13}C_6$-glucose in glycolysis and TCA cycle intermediates in rod photoreceptor-specific *Gls* conditional knockout (cKO) mice compared to wild-type (WT) mice.

**Figure supplement 2.** Mass isotopologue distribution of $^{13}C_5$-Gln in TCA cycle intermediates in rod photoreceptor-specific *Gls* conditional knockout mice (cKO) compared to wild-type (WT) mice.

cycle intermediates was not observed in the cKO retina (*Figure 5A*, *Figure 5—figure supplement 1*). Hence, it is unlikely rod PRs are increasing glucose oxidation to maintain the levels of TCA cycle intermediates and mitochondrial function.

## *Gls* cKO in rod photoreceptors decreases the utilization of Gln in the TCA cycle

We initially anticipated that GLS knockout in rod PRs would significantly alter the relative TCA cycle metabolite pools in the cKO retina, similar to what is seen in Gln-dependent cancer cells (*Daemen et al., 2018*; *Yang et al., 2017*), as PRs and cancer cells have common metabolic relationships (*Du et al., 2013b*; *Ng et al., 2015*; *Rajala, 2020*), and previous ex vivo studies demonstrated Gln can supplement the TCA cycle in the retina (*Grenell et al., 2019*; *Tsantilas et al., 2021*). However, as observed in *Figure 4F*, very few changes were observed in the relative pool sizes of TCA cycle metabolites in the cKO compared to WT retina. To delineate how Gln is metabolized, we used LC-MS/MS to trace the metabolic fate of uniformly-labeled $^{13}C_5$-Gln in the WT and cKO retina in vivo at P14. Uniformly-labeled $^{13}C_5$-Gln was intraperitoneally injected into the mice and the retina harvested 45 min later. In accordance with GLS being the enzyme that initiates glutaminolysis (*Yang et al., 2017*), the fractional labeling of glutamate, TCA cycle intermediates, and pyruvate was decreased in the P14 cKO retina as compared to WT (*Figure 5C,D*, *Figure 5—figure supplement 2*). Previous ex vivo studies have also observed that isotopically labeled $^{13}C_5$-Gln can contribute to m+3 pyruvate in the retina possibly via the decarboxylation of m+4 malate by malic enzyme (*Grenell et al., 2019*).

a-ketoglutarate (α-KG) is the main entry point for Gln into the TCA cycle (*Figure 5C*) and has been shown to be a key metabolite in Gln metabolism (*Yang et al., 2017*). In vivo stable isotope tracing demonstrated decreased incorporation of Gln carbons into α-KG in the cKO retina compared to the WT (*Figure 5D*). Furthermore, supplementation with α-KG has been shown to improve PR survival in mouse models of inherited retinal disease (*Wert et al., 2020*). So, we attempted to rescue the PR degeneration phenotype in the GLS cKO mouse by supplementing cKO animals with 10 mg/mL of α-KG in the drinking water starting at P4 (*Rowe et al., 2021*; *Wert et al., 2020*). A small, but significant increase in ONL thickness was identified in α-KG-treated animals at P22 using OCT (*Figure 5E*). These data further suggest that Gln's role in supporting the TCA cycle is not the major mechanism by which PRs utilize Gln to suppress PR apoptosis.

## *Gls* cKO retina has reduced NEAA levels and an upregulated integrated stress response

Gln and GLS-derived Glu play a central role in the biosynthesis of several NEAAs (*Yoo et al., 2020*). So, the disruption of GLS-initiated Gln catabolism may be causing a disruption in available NEAAs for biomass production. To this end, targeted metabolomics on the cKO retina at P14 showed a significant increase in the substrate of GLS, Gln, and a significant decrease in its NEAA product Glu, further illustrating loss of GLS function (*Figure 6A and B*). The NEAA Asp was also significantly reduced (*Figure 6A*), which is consistent with previous studies that inhibited GLS in cancer cells (*Alkan et al., 2018*; *Okazaki et al., 2017*). Notably, similar decreases in Glu and Asp were noted in IND-cKO compared to WT retina at a point prior to PR degeneration (*Figure 6—figure supplement 1*).

Deprivation of amino acids can activate the integrated stress response (ISR), which responds to an array of stressors to maintain cellular homeostasis and particularly, protein homeostasis. Acutely, ISR activation can protect cells by temporarily halting protein synthesis. However, chronic ISR activation and global protein synthesis inhibition can trigger apoptosis (*Pakos-Zebrucka et al., 2016*). ISR activation leads to phosphorylation of the alpha subunit of eukaryotic translation initiation factor 2 (eIF2α), which reduces global protein synthesis while preferentially allowing for the translation of activating transcription factor 4 (ATF4; *Pakos-Zebrucka et al., 2016*). Considering the reduced levels of NEAAs Glu and Asp, retinas from P14 GLS cKO animals were assayed for these ISR components using Western blotting (*Figure 6C and D*), which demonstrated increased levels of both phosphorylated eIF2α and total ATF4, suggesting ISR activation. To determine if global protein synthesis was affected in GLS cKO retinas, we applied the SUnSET (SUrface SEnsing of Translation) method to quantify protein synthesis in WT and cKO retinas at P14 (*Fort et al., 2022*). These data show a significant decrease of puromycin incorporation into nascent polypeptide chains, indicating a decrease in global protein synthesis consistent with ISR activation (*Figure 6E and F*). To determine if PR degeneration could be

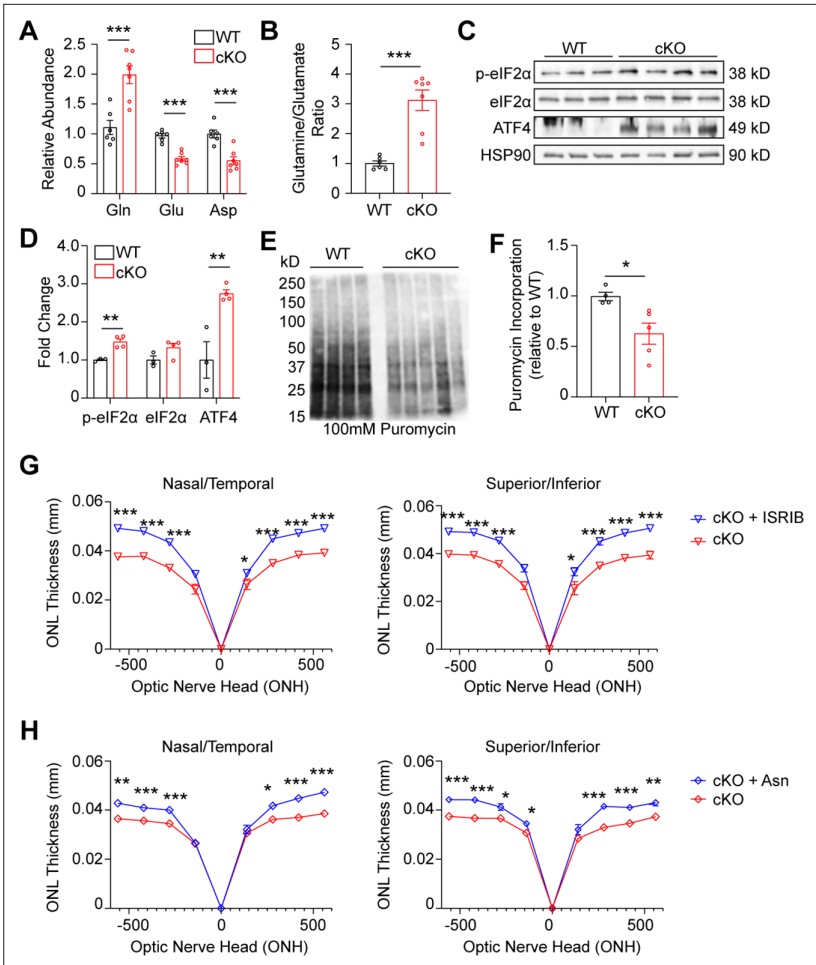

**Figure 6.** GLS cKO retina demonstrates decreased nonessential amino acids, ISR activation, and decreased global protein synthesis. (**A**) Amino acids significantly altered in the *Gls* cKO mouse retina at P14. Relative abundance is the ion intensity relative to WT retina. N=6–7 animals per group. (**B**) Ratio of glutamine to glutamate in WT and cKO retina. N=6–7 animals per group. (**C**) Western blot of ISR proteins phospho-eIF1α$^{S51}$ (p-eIF2α), total eIF2α and ATF4 in WT and cKO mice. N=3–4 animals per group. (**D**) Quantitation of western blot in Panel C. (**E**) Western blot of protein puromycinylation in the WT and cKO mouse retina at P14 harvested 30 min after systemic puromycin administration. (**F**) Quantitation of puromycin incorporation in WT and cKO retina. N=4–5 animals per group. (**G**) ONL thickness at P21 in cKO mice as assessed by OCT following intraperitoneal injection of ISRIB (2.5 mg/kg) or vehicle (50% PEG 400, 43.4% saline, 6.6% DMSO) from P5-P21. N=3–6 animals per group. (**H**) ONL thickness at P21 in cKO mice as assessed by OCT following intraperitoneal injection of Asn (200 mg/kg) or vehicle (PBS) from P5-P21. N=3–5 animals per group. Statistical differences in (**A**), (**B**), (**D**), (**F**), (**G**), and (**H**) are based on an unpaired two-tailed Student's t-test where *p<0.05, **p<0.01 and ***p<0.001. Data are presented as mean ± SEM. Gln: glutamine, Glu: glutamate, Asp: aspartate, ONL: outer nuclear layer.

The online version of this article includes the following source data and figure supplement(s) for figure 6:

**Source data 1.** Original western blot membranes corresponding to *Figure 6C and E*.

**Source data 2.** Unannotated western blot membranes corresponding to *Figure 6C and E*.

**Figure supplement 1.** Amino acid levels are altered in the IND-cKO retina.

**Figure supplement 2.** ISRIB does not impact retinal anatomy in WT mice and prolongs PR survival in the cKO retina.

**Figure supplement 3.** Histology confirms ISRIB and Asn prolong photoreceptor survival in the GLS cKO retina.

rescued by inhibiting the ISR, ISRIB (2.5 mg/kg) was administered systemically daily starting at P5. In WT mice, ISRIB treatment did not impact retinal anatomy as assessed by OCT at P21 (*Figure 6—figure supplement 2A*). Notably, at P21, ISRIB treatment in the cKO mouse significantly increased ONL thickness compared to vehicle using OCT (*Figure 6G*) and histology (*Figure 6—figure supplement 3A*), and this effect was sustained at P28 (*Figure 6—figure supplement 2B*). These results suggest that ISR activation is downstream of Gln catabolism and contributes to PR degeneration in the cKO mouse retina.

Previous work has demonstrated that when certain cancer cell lines are faced with Gln deprivation, α-KG alone does not restore cell proliferation (*Pavlova et al., 2018*; *Son et al., 2013*). Rather, NEAA supplementation is necessary to restore proliferation in these reports. Similarly, α-KG supplementation had an incomplete survival impact on PR survival, showing approximately 10% increase in ONL thickness in cKO animals (*Figure 5E*). Since systemic or intravitreal glutamate supplementation has demonstrated retinal toxicity (*Sisk and Kuwabara, 1985*), and Asp concentration is normally low in plasma with poor cellular membrane permeability (*Sullivan et al., 2018*), we investigated other NEAA approaches to rescue the PR degeneration observed in the GLS cKO mouse. The potential of asparagine (Asn) to rescue PR degeneration was chosen due to the growing list of Asn-mediated cellular processes that improve cancer cell survival and growth under metabolic stress, such as Gln deprivation. These Asn-mediated cellular processes include inhibiting ISR-induced apoptosis (*Zhang et al., 2014*),(1) promoting protein synthesis (*Pavlova et al., 2018*), and maintaining Asp pools (*Halbrook et al., 2022*). All these processes may be relevant to the PR degeneration observed in the cKO mouse. Following administration of systemic Asn (200 mg/kg) daily starting at P5, in vivo analysis of retinal structure via OCT and histology at P21 demonstrated a significant improvement in ONL thickness compared to animals treated with vehicle, with a 22–26% increase in ONL thickness in some retinal locations (*Figure 6*, *Figure 6—figure supplement 3*).

## Discussion

In this study, we demonstrate for the first time the importance of Gln catabolism in PR metabolism, function, and survival in vivo. Rod PRs lacking GLS demonstrated rapid and complete degeneration with concomitant functional loss. Mechanistically, lack of GLS-initiated Gln catabolism in rod PRs, specifically, resulted in changes in redox homeostasis and a decrease in the fractional labeling of TCA cycle intermediates from stable isotope-labeled Gln in vivo. However, Gln anaplerosis did not appear to be driving the TCA cycle as targeted metabolomics showed few changes in the relative levels of TCA cycle intermediates, and supplementation with αKG showed a modest rescue effect in PR degeneration. The NEAAs Glu and Asp were found to be significantly decreased in the cKO mouse retina, and in accordance with this AA deprivation, the ISR was upregulated with a reduction in global protein synthesis suggesting Gln catabolism in rod PRs plays a significant role in supporting biomass production via NEAA synthesis. Interestingly, supplementing cKO mice with Asn significantly rescued PR degeneration, revealing a novel link between Gln and Asn metabolism in PRs. This work further demonstrates that PRs have the flexibility to utilize fuel sources other than glucose to meet their metabolic needs and that Gln is a critical amino acid that supports PR cell biomass, redox balance, and survival.

The rapid degenerative phenotype observed with conditional deletion of *Gls* in rod PRs was unexpected considering glucose is viewed as the primary substrate for the retina and PRs (*Swarup et al., 2019*). The *Gls* cKO mouse demonstrated comparable ONL thickness to WT controls at P14 with approximately 50% loss of PR cell bodies by P21 (*Figure 1—figure supplement 3A–B*) and near complete PR degeneration by P84 (*Figure 1C*). In contrast, conditional deletion of numerous enzymes within central glucose metabolism, including hexokinase 2 (HK2; *Weh et al., 2020*), pyruvate kinase M2 (PKM2; *Wubben et al., 2017*), glucose transporter 1 (GLUT1; *Daniele et al., 2022*), mitochondrial pyruvate carrier 1 (MPC1; *Grenell et al., 2019*), and lactate dehydrogenase A (LDHA; *Rajala et al., 2023*), results in a slower, age-related degeneration. For example, loss of GLUT1 in the retina and rod PRs does not demonstrate 50% thinning of the ONL until approximately 4 months of age (*Daniele et al., 2022*), and the others noted above demonstrate even slower outer retinal degeneration. It has also been shown that PRs can utilize fatty acid β-oxidation for energy (*Joyal et al., 2016*). Interestingly, in retinas lacking the very-low-density lipoprotein receptor (VLDLR), which facilitates fatty acid uptake, the PRs remained largely intact. In any of these transgenic mouse models, PRs may use other

transporters to take up fatty acids or glucose, rewire their metabolism, or utilize metabolic redundancies to maintain metabolic homeostasis and stave off degeneration (*Subramanya et al., 2023*; *Wubben et al., 2017*). Our data show that any metabolic reprogramming that is occurring in the cKO mouse retina appears unable to significantly circumvent the significant and rapid PR degeneration suggesting the importance of Gln catabolism in rod PRs. Furthermore, inducing GLS knockdown in mature PRs also demonstrated rapid PR degeneration (*Figure 3*).

Traditionally, it has been thought that Gln maintains cell survival through Gln-derived α-KG and maintenance of the TCA cycle (*Zhang et al., 2014*).(1) While stable isotope tracing with uniformly labeled Gln demonstrated a reduction in fractional labeling of the TCA cycle intermediates (*Figure 5D*), targeted metabolomics showed few changes in the relative levels of the intermediates (*Figure 4F*) and supplementation with αKG did not rescue PR degeneration to a large degree (*Figure 5E*). Gln is also required for the biosynthesis of NEAAs, and the NEAAs Glu and Asp were reduced nearly two-fold in the retina of cKO mouse (*Figure 6A*). Glu is the product of the GLS reaction so it is not surprising that Glu was substantially reduced. Glu is involved in glutathione biosynthesis directly and indirectly, can be converted into α-KG to enter the TCA cycle, and is crucial for the biosynthesis of NEAAs including Asp (*Yoo et al., 2020*). Alterations in redox homeostasis were observed in the retina of the cKO mouse in accordance with the reduced level of Glu (*Figure 4C-E*). While these redox imbalances likely contribute to the PR degeneration observed, it is unlikely these small but statistically significant changes fully account for the rapid and complete PR degeneration. Future rescue studies with antioxidants, such as N-acetylcysteine, are needed to shed light on the role of redox imbalance in this novel transgenic mouse model.

Previous ex vivo studies demonstrated that Gln can contribute carbons to Asp synthesis via the TCA cycle and/or contribute a nitrogen via glutamate-oxaloacetate transaminases (*Du et al., 2013a*; *Xu et al., 2020*), the latter of which have been shown to be critical in the Gln metabolic rewiring of cancer cells (*Son et al., 2013*) and more recently, in rod PR metabolism, function, and survival (*Subramanya et al., 2023*). Our data shows that Gln-derived carbons via Glu are entering the TCA cycle in vivo and contributing to the synthesis of Asp (*Figure 5*), and this contribution is reduced in the cKO mouse retina. Based on the previous studies noted above, Gln may also be contributing to Asp synthesis via the glutamate-oxaloacetate transaminase catalyzed transfer of a nitrogen from Glu. Future studies using amine-labeled Gln are needed to dissect the contributions of Gln-derived carbon versus nitrogen to the synthesis of Asp in the retina in vivo.

Asp is a proteinogenic amino acid that has many biosynthetic roles (*Alkan et al., 2018*), and ex vivo neural retina studies have suggested that the retina needs Asp to maintain its metabolic homeostasis (*Li et al., 2020*). In accordance with this data, Asp was one of the few metabolites that was significantly reduced in the retina of cKO mouse prior to rapid PR degeneration (*Figure 6A*). Based on ex vivo studies, it has been postulated that Asp utilization in the retina is necessary to maintain aerobic glycolysis and mitochondrial metabolism through the recycling of NADH to $NAD^+$, shuttling electrons into the mitochondria via the malate-aspartate shuttle, replenishing oxaloacetate for biosynthesis and/or producing pyruvate via malic enzyme (*Li et al., 2020*). Our in vivo metabolomics data further support this last point as uniformly labeled Gln contributed to m+3 pyruvate in the retina possibly via the decarboxylation of m+4 malate by malic enzyme (*Figure 6D*). The m+4 isotopologue of malate could arise from m+4 Asp via glutamate-oxaloacetate transaminase (GOT) and malate dehydrogenase (MDH) as part of the malate-aspartate shuttle. Furthermore, the decarboxylation of malate by malic enzyme produces pyruvate and also NADPH. A previous study revealed high malic enzyme activity in the retina and suggested it is capable of producing NADPH in the retina (*Winkler et al., 1986*). The observation that malate was reduced (*Figure 4F*) and the $NADP^+$/NADPH ratio was increased (*Figure 4C*) in the cKO retina further supports a metabolic pathway in PRs where the Gln carbon skeleton is converted into Asp, then OAA, malate, and finally, pyruvate via the cytosolic enzymes of the malate-aspartate shuttle and malic enzyme. This non-canonical Gln metabolism pathway is also necessary to sustain cancer cell growth (*Son et al., 2013*).

The m+4 isotopologue of oxaloacetate (OAA) can also be converted to m+3 pyruvate by phosphoenolpyruvate carboxykinase (PEPCK). A recent ex vivo study postulated that PRs can utilize different metabolic cycles, such as the Cahill cycle or mini-Krebs cycle, to uncouple glycolysis from oxidative phosphorylation (*Chen et al., 2024*). These cycles are fueled by Gln and require PEPCK to replenish pyruvate from OAA. While the lack of m+4 citrate in both the P14 WT and cKO retina and decrease in

m+2 citrate in the cKO retina (*Figure 5—figure supplement 2*) may further support the existence of these previously postulated metabolic cycles, which avoid the citrate synthase step of the TCA cycle as well as others, further studies are needed to confirm the activity of these pathways in the retina and specifically, the PRs.

Interestingly, we did not observe any significant changes in glycolysis or mitochondrial function in the cKO retina despite a reduction in the relative level of Asp. It is hypothesized that Asp catabolism sustains glycolysis and oxidative phosphorylation in the retina by recycling NADH to $NAD^+$ and shuttling electrons into the mitochondria via the malate-aspartate shuttle (*Li et al., 2020*). However, in the GLS cKO retina, the $NAD^+$/NADH ratio (p=0.125) and its oft-used proxy, the pyruvate/lactate ratio (p=0.192), were not statistically significantly altered. Compensatory metabolic rewiring, which has been seen in other conditional knockout mouse models using the same Cre-recombinase system (*Subramanya et al., 2023*; *Wubben et al., 2017*), may be maintaining the $NAD^+$/NADH ratio in the cKO retina at P14 when there are equal numbers of PRs in the WT and cKO retina. To this end, the expression of the *Mdh1* gene, which is a component of the malate-aspartate shuttle that recycles NADH to $NAD^+$ in the cytoplasm, was upregulated in the cKO retina (*Figure 4—figure supplement 1*). The conversion of dihydroxyacetone phosphate (DHAP) to glycerol 3-phosphate via glycerol 3-phosphate dehydrogenase is an alternative pathway to regenerate $NAD^+$. DHAP levels were significantly reduced in the cKO retina possibly suggesting increased activity of this pathway (*Figure 4F*). Yet, the gene expression and activity of glycerol 3-phosphate dehydrogenase in the retina has been shown to be low (*Adler and Klucznik, 1982*; *Voigt et al., 2020*).

Asp is also the immediate precursor to Asn. Intracellular levels of Asn are the lowest among the NEAAs (*Zhang et al., 2014*), so it was not surprising that Asn was below the level of confident detection in our targeted metabolomics analyses on the WT or cKO retina. Asn had been shown to rescue cancer cells from ISR-induced apoptosis, increase protein synthesis during Gln deprivation, and protect the Asp pool (*Halbrook et al., 2022*; *Pavlova et al., 2018*; *Zhang et al., 2014*). As metabolic similarities exist between PRs and cancer cells (*Du et al., 2013b*; *Ng et al., 2015*), and the *Gls* cKO mouse retina demonstrated ISR activation and apoptosis (*Figures 1G-I and 6C-D*), decreased protein synthesis (*Figure 6E*), and decreased Asp (*Figure 6A*), Asn supplementation was explored and proved effective as observed in the rescue of PR degeneration in the cKO mouse (*Figure 6H*). These results offer the first evidence for a role of Asn downstream of Gln metabolism in PR survival, but further studies are necessary to define which of the Asn-mediated processes is crucial for PR neuroprotection following targeted *Gls* knockout. Of note, while Asn supplementation provided a greater PR neuroprotective effect than α-KG, the two supplements had different routes of administration with Asn being provided intraperitoneally as previously described (*Xu et al., 2021*) and α-KG being provided in the drinking water as had previously improved PR survival in a mouse model of inherited retinal disease (*Wert et al., 2020*). It is unclear if supplementing α-KG to GLS cKO animals intraperitoneally would further boost retinal protection. Yet, the relative abundance of α-KG and most other TCA cycle intermediates were unchanged between WT and cKO retinas suggesting Gln may not be driving the TCA cycle in PRs.

The ISR is activated with reduced global protein synthesis in the *Gls* cKO mouse retina (*Figure 6C-F*) and inhibiting the ISR with ISRIB delayed PR degeneration (*Figure 6G*). ISR activation is a hallmark of neurodegenerative diseases including retinal degenerative diseases (*Gorbatyuk et al., 2020*). Chronic activation of the ISR and protein synthesis attenuation has been observed in a multitude of preclinical models of retinal degeneration and shown to contribute to PR degeneration (*Gorbatyuk et al., 2020*). Metabolic dysfunction and deprivation of key nutrients, such as glucose and amino acids, are not only known stressors that activate the ISR (*Pakos-Zebrucka et al., 2016*), but also underlying mechanisms of PR degeneration (*Aït-Ali et al., 2015*; *Caruso et al., 2020*; *Pan et al., 2021*). There is a paucity of studies that examine the link between metabolism and the ISR in retinal degenerative disease, which is a critical gap in our knowledge since identification of molecular and metabolic pathways triggering PR death is likely to reveal novel targets for therapeutic intervention. A recent study demonstrated that imbalanced pro-apoptotic ISR signaling contributes to deoxysphingolipid-mediated toxicity in retinal disease and identified potential therapeutic strategies, such as pharmacologic enhancement of ATF6 activity or treatment with ATF6-regulated neurotrophic factor MANF, that attenuate the associated retinal degeneration (*Rosarda et al., 2023*). The novel transgenic mouse model of retinal degeneration described here provides a unique tool to obtain further insight on the nexus of metabolism,

ISR activation, and protein synthesis attenuation in PR degeneration to identify pharmacologically and metabolically tractable nodes for therapeutic intervention.

Collectively, our results indicate that Gln is critical for maintaining the pools of key biosynthetic precursors, Glu and Asp, in rod PRs and disrupting Gln catabolism results in profound loss of PR function and survival in part secondary to an imbalance in ISR activation and protein synthesis attenuation. Glucose remains central in PR metabolism, but improving our understanding of other metabolic pathways that support PR function and survival and how these metabolic pathways connect with cell death mechanisms could be transformative for preventing PR degeneration and vision loss in a multitude of retinal degenerative diseases.

## Ideas and speculation

Beyond glucose, the metabolic pathways integral to photoreceptor health remain largely unknown. This is a critical knowledge gap as identification of these pathways is likely to reveal new strategies for therapeutic intervention. This work demonstrates that rod photoreceptors depend on glutamine catabolism and suggests a metabolic axis where glutamine catabolism in rod photoreceptors supports the production of aspartate and asparagine to promote anabolism and prevent signaling through the pro-apoptotic ISR pathway. Considering activation of the ISR is a hallmark of neurodegenerative diseases and metabolic dysfunction underlies photoreceptor degeneration, defining the pathways by which glutamine catabolism contributes to photoreceptor health is likely to identify nodes that may be targeted to make photoreceptors less vulnerable to stress.

## Key resources table

| Reagent type (species) or resource | Designation | Source or reference | Identifiers | Additional information |
|---|---|---|---|---|
| Genetic reagent (*Mus musculus*) | *Gls^fl/fl* | PMID:26778975 | RRID:IMSR_JAX:017894 | |
| Genetic reagent (*Mus musculus*) | *Rho^Cre* | PMID:16636658 | RRID:IMSR_JAX:032909 | |
| Genetic reagent (*Mus musculus*) | *Gls^fl/fl;Rho^Cre+* | This study | | |
| Genetic reagent (*Mus musculus*) | *Pde6g^Cre:ERT2* | PMID:26301813 | | |
| Genetic reagent (*Mus musculus*) | *Gls^wt/wt;Pde6g^Cre:ERT2* | This study | | |
| Genetic reagent (*Mus musculus*) | *Gls^fl/fl;Pde6g^Cre:ERT2* | This study | | |
| Antibody | (Mouse monoclonal) anti-GLS | Proteintech | 66265–1-Ig | 1:200; immunofluorescence |
| Antibody | (Rabbit polyclonal) anti-GLS2 | Abcam | Ab113509 | 1:200; immunofluorescence |
| Antibody | (Mouse monoclonal) anti-Rhodopsin | Abcam | Ab5417 | 1:1000; immunofluorescence |
| Antibody | (Goat polyclonal) anti-OPN1MW/LW | Santa Cruz Biotechnology | Sc-22117 | 1:200; immunofluorescence |
| Antibody | (Mouse monoclonal) anti-BRN3A | Santa Cruz Biotechnology | Sc-8429 | 1:100; immunofluorescence |
| Antibody | (Rabbit polyclonal) anti-calretinin | Millipore Sigma | C7479 | 1:100; immunofluorescence |
| Antibody | (Rat monoclonal) anti-GFAP | Thermo Fisher | 13–300 | 1:200; immunofluorescence |
| Antibody | (Mouse monoclonal) anti-Chx10 | Santa Cruz Biotechnology | Sc-365519 | 1:200; immunofluorescence |
| Antibody | (Mouse monoclonal) anti-Bassoon | Enzo | SAP7F407 | 1:1000; immunofluorescence |
| Antibody | (Goat polyclonal) anti-mouse Alexa 488 | Invitrogen | A11001 | 1:1000; immunofluorescence |
| Antibody | (Donkey polyclonal) anti-mouse Alexa 594 | Jackson ImmunoResearch Laboratories | 715-585-151 | 1:500, immunofluorescence |
| Antibody | (Donkey polyclonal) anti-rabbit Alexa 594 | Jackson ImmunoResearch Laboratories | 711-585-152 | 1:500, immunofluorescence |
| Antibody | (Donkey polyclonal) anti-goat Alexa 647 | Invitrogen | A21447 | 1:2000; immunofluorescence |

*Continued on next page*

*Continued*

| Reagent type (species) or resource | Designation | Source or reference | Identifiers | Additional information |
|---|---|---|---|---|
| Antibody | (Donkey polyclonal) anti-rat Alexa 594 | Invitrogen | A21209 | 1:500; immunofluorescence |
| Antibody | (*Arachis hypogea*) Lectin PNA Alexa 594 conjugate | Invitrogen | L32459 | 1:200; immunofluorescence |
| Antibody | (Wheat germ) Agglutinin (WGA) Alexa 594 conjugate | Invitrogen | W11262 | 1:1000; immunofluorescence |
| Antibody | (Rabbit polyclonal) anti-KGA-specific GLS | Proteintech | 20170–1-AP | 1:1000; western |
| Antibody | (Rabbit polyclonal) anti-GAC-specific GLS | Proteintech | 19959–1-AP | 1:1000; western |
| Antibody | (Rabbit polyclonal) anti-GLS | Proteintech | 12855–1-AP | 1:1000; western |
| Antibody | (Rabbit polyclonal) anti-VDAC | Cell Signaling Technology | 4866 | 1:1000; western |
| Antibody | (Mouse monoclonal) Anti-TIM23 | BD Biosciences | 611223 | 1:1000; western |
| Antibody | (Mouse monoclonal) anti-HSP90 | Cell Signaling Technology | 4877 | 1:2000; western |
| Antibody | (Horse polyclonal) anti-mouse-HRP-linked | Cell Signaling Technology | 7076 | 1:5000; western |
| Antibody | (Goat polyclonal) anti-rabbit-HRP-linked | Cell Signaling Technology | 7074 | 1:5000; western |
| Antibody | (Rabbit monoclonal) anti-eIF2α | Cell Signaling Technology | 5324 | 1:1000; western |
| Antibody | (Rabbit monoclonal) anti-Phospho-eIF2α | Cell Signaling Technology | 3398 | 1:5000; western |
| Antibody | (Rabbit polyclonal) anti-ATF4 | Invitrogen | PA5-27576 | 1:1000; western |
| Antibody | (Mouse monoclonal) anti-puromycin | Biolegend | 381502 | 1:1000; western |
| Antibody | (Mouse monoclonal) Total OXPHOS rodent antibody | Abcam | Ab110413 | 1:1000; western |

## Materials and methods

### Animals

All animals were treated in accordance with the Association for Research in Vision and Ophthalmology Statement for the Use of Animals in Ophthalmic and Vision Research. The protocol was approved by the University Committee on Use and Care of Animals of the University of Michigan (PRO00011133). All animals were housed under standard husbandry conditions at room temperature in 12 hr light/12 hr dark cycles unless explicitly stated in the text. Both male and female mice were used for all experiments. A transgenic mouse where *Gls* is selectively deleted from rod photoreceptors was created by crossing mice with Lox-P sites flanking exon 1 of the *Gls* gene (*Gls*$^{fl/fl}$, courtesy of Dr. Stephen Rayport, Columbia University) with *Rho*$^{Cre}$ mice, in which Cre-recombinase expression is driven specifically in rod PRs. *Gls*$^{fl/fl}$ and *Rho*$^{Cre}$ mice have been previously described (*Le et al., 2006*; *Mingote et al., 2015*). Animals were screened for the *rd8* mutation (*Mattapallil et al., 2012*). Alpha-ketoglutarate (α-KG, Millipore-Sigma, St. Louis, MO, USA, Cat # K1128) was provided to mice in their drinking water (10 mg/mL) starting at P4. Asparagine (200 mg/kg, Millipore-Sigma, Cat # A4159) or vehicle (PBS) was injected IP starting at P5. ISRIB (2.5 mg/kg, Caymen Chemical, Ann Arbor, MI, USA, Cat # 16258) or vehicle (50% PEG 400, 43.4% saline, 6.6% DMSO) (*Halliday et al., 2015*) was injected IP starting at P5. For inducible deletion of *Gls*, *Gls*$^{fl/fl}$ mice were crossed to *Pde6g*$^{Cre:ERT2}$ mice (courtesy of Dr. Stephen Tsang, Columbia University) (*Koch et al., 2015*). The Cre-recombinase was activated via IP injection of tamoxifen (Millipore-Sigma, Cat # T5648) at a concentration of 100 mg/kg bodyweight for 5 consecutive days. Whole retinas were extracted from animals using the 'cut-and-pick' method as previously described (*Winkler, 1981*), being careful to remove any adherent ciliary body or RPE before processing. Total retina was then either snap frozen on dry ice (Western blotting, metabolomics), immersed in RNAlater (QIAGEN, Hilden, Germany, Cat # 76104) for qRT-PCR, or used immediately (BaroFuse or NADP$^+$/ NADPH bioluminescent assay).

## In vivo functional and structural assessment

Visual function was assessed as previously described (*Weh et al., 2020*; *Wubben et al., 2017*). Mice were anesthetized using a mixture of ketamine/xylazine (90/10 mg/kg) and their pupils were dilated using 1% tropicamide and 2.5% phenylephrine ophthalmic drops. Retinal function was determined using a Diagnosys Celeris ERG system (Diagnosys LLC, Lowell, MA, USA) following overnight dark adaptation. In vivo retinal thickness was measured using the Envisu-R SD-OCT imager (Leica Microsystems Inc, Buffalo Grove, IL, USA). A 1.5 mm horizontal B-scan (1000 A-scans × 100 frames) and a 1.5 mm × 1.5 mm rectangular volume (1000 A-scans × 36 B-scans × 6 frames) were captured, registered and averaged using the built-in software, and analyzed using the Diver 1.0 software suite (Leica Microsystems). Images were segmented manually to determine total retinal, outer nuclear layer, and combined inner segment/outer segment thickness. Measurements were taken at 15 points on a 9 × 9 grid and averaged as previously described (*Weh et al., 2022*).

## Immunofluorescence

Mouse eyes were enucleated and immersed in 4% paraformaldehyde overnight before embedding in paraffin and sectioned at 6 mm thickness. Following standard protocols, sections were de-paraffinized and antigen retrieval performed as previously described (*Weh et al., 2020*; *Wubben et al., 2017*). Sections were blocked with 1% bovine serum albumin (BSA, Millipore-Sigma, Cat # A9647) in 1 X phosphate buffered saline (PBS, Thermo Fisher Scientific, Waltham, MA, USA, Cat # BP399) with 0.125% Tween 20 (Thermo Fisher Scientific, Cat # BP337) and 10% normal goat serum prior to incubating with primary antibody in 1% BSA and 1% normal goat serum overnight at 4 °C. Slides were then washed, secondary antibody applied for 1 hr at room temperature before washing, and finally, counterstained with DAPI (Thermo Fisher Scientific, Cat # P36930). Images were obtained on a Leica DM6000 microscope with a 40 X objective. The antibodies used for immunofluorescence are found in the key resources table.

## TUNEL staining and cell counts

TUNEL staining was performed as previously described using the DeadEnd kit (Promega, Madison, WI, USA, Cat # G3250; *Wubben et al., 2017*; *Wubben et al., 2020*). TUNEL-positive cells were counted in a masked fashion and normalized to the total number of nuclei using a custom ImageJ macro (*Busov and Besirli, 2014*; *Wubben et al., 2017*; *Wubben et al., 2020*). Tissue sections through the plane of the optic nerve were also stained with hematoxylin and eosin and the total number of nuclei in the ONL were determined after normalization to inner retinal area (*Wubben et al., 2020*).

## Sub-cellular fractionation

Whole retina was fractionated into cytosolic and post-cytosolic (mitochondrial enriched) fractions as previously described (*Weh et al., 2020*) using the Subcellular Protein Fractionation Kit for Tissues (Thermo Fisher Scientific, Cat # 87790). Both retinas from a single animal were pooled and homogenized with a Dounce homogenizer in cytoplasmic extraction buffer supplemented with protease (Halt Protease Inhibitor Cocktail, Thermo Fisher Scientific, Cat # 87786) and phosphatase (Halt Phosphatase Inhibitor Cocktail, Thermo Fisher Scientific, Cat # 78420) inhibitors. The retinal lysate was then centrifuged at 10,000 x relative centrifugal force (RCF) for 10 min at 4 °C. The resulting supernatant was saved as the cytosolic fraction, and the resulting pellet was resuspended in RIPA lysis buffer (Thermo Fisher Scientific, Cat # 89900) that included protease and phosphatase inhibitors (Cell Signaling Technology, Danvers, MA, USA, Cat # 5872) and sonicated at 20% amplitude with 1 s on/off pulse for 10 s. The lysate was centrifuged for 10 min at 10,000 x RCF at 4 °C. The resulting supernatant was saved as the mitochondrial enriched fraction. The percentage of GLS in each fraction was determined using Western blotting.

## Western blotting

Immunoblots were performed as previously described (*Weh et al., 2020*). Protein estimation was performed using the Pierce BCA kit (Thermo Fisher Scientific, Cat # 23225). Equivalent micrograms of protein from each sample were diluted using 4 X Laemmli buffer (Bio-Rad, Hercules, CA, USA, Cat # 1610747) supplemented with β-mercaptoethanol (Millipore-Sigma, Cat # M6250) before heating at 95 °C for 5 min and finally loaded onto 4–20% polyacrylamide gel (Bio-Rad, Cat # 4561094). Samples

were then transferred to a PVDF membrane using the Trans-Blot Turbo Transfer System (Bio-Rad, Cat # 1704150). Membranes were blocked using 5% non-fat milk powder diluted in TBST (Tris-buffered Saline, Bio-Rad, Cat # 1706435, supplemented with 0.1% Tween-20, Thermo Fisher Scientific, Cat # 28320) for 4 hr at room temperature. Primary antibodies were diluted in 5% BSA and added to blots before incubating overnight at 4 °C. Blots were then washed and appropriate secondary antibody was added for 1 hr at room temperature. Chemiluminescence was performed using the SuperSignal West Dura/Femto Extended Duration Substrate (Thermo Fisher Scientific, Cat # 34075 and 34094) and the immunoblots were imaged with an Azure 600 imaging system (Azure Biosystems; Dublin, CA USA). All antibodies and dilutions used are found in the key resources table.

## Quantitative real-time PCR

Total RNA was extracted from whole retina using the RNeasy Mini Kit (QIAGEN, Cat # 74104) following the manufacturer's protocol. Isolated RNA was assayed for quantity and quality with a Nanodrop 1000 (Thermo Fisher Scientific) and 1 µg of RNA was used as input for cDNA synthesis using the RNA QuantiTect transcription kit (Qiagen, Cat # 205311). Approximately 100 ng of cDNA was used as a template for each qRT-PCR reaction using the PowerTrack SYBR Green supermix (Thermo Fisher Scientific, Cat # 46109) as previously described (*Subramanya et al., 2023*). The Ct values for *Actb* were used to determine relative transcript expression levels using the $2^{-\Delta\Delta Ct}$ method with a cycle threshold cutoff of 35 cycles for the presence of transcript. The geometric mean was used to normalize samples. Custom qRT-PCR primers (*Supplementary file 1*) designed to specifically detect spliced transcripts were used to determine transcript levels following the above protocol.

## Metabolomics

For unlabeled targeted metabolomics, both retinas from a single animal were rinsed in PBS, combined, and snap-frozen on dry ice. Metabolites were extracted using 80% methanol at –80 °C and an OMNI Bead Ruptor (OMNI International, Kennesaw, GA, USA, Cat # 19–050 A). Lysates were centrifuged at 14,000 x RCF for 10 min at 4 °C, and the supernatant stored at –80 °C until being processed in the SpeedVac. The pellet from each sample was saved for protein estimation. To determine the protein concentration for each sample, 150 µL of 0.1 M NaOH was added to the pellet for 24 hr at 37 °C. The sample was then vortexed and centrifuged at 5000 x RCF at room temperature. The protein estimation was performed as described above. The protein concentration of each sample was used to normalize amount of sample for lyophilization with a SpeedVac concentrator (Thermo Fisher Scientific, Cat # 13875355). Dried metabolite pellets were resuspended for liquid chromatography-coupled mass spectrometry (LC/MS) analysis using an Agilent Technologies Triple Quad 6470 instrument (Santa Clara, CA, USA) as previously described (*Wubben et al., 2020*). Previously published parameters were used for data collection (*Yuan et al., 2012*). Agilent MassHunter Workstation Quantitative Analysis Software (B0900) was used to process raw data. Additional statistical analyses were performed in Microsoft Excel. Each sample was normalized by the total intensity of all metabolites to reflect sample protein content. To obtain relative metabolites, each metabolite abundance level in each sample was divided by the mean of the abundance levels across all control samples.

To analyze the incorporation of non-radioactive stable isotope carbon-13 ($^{13}$C) into metabolites in central carbon metabolism and related pathways, mice were intraperitoneally injected with 2 g/kg of uniformly labeled $^{13}$C$_6$-glucose (Cambridge Isotope Laboratories, Cat # CLM-1396) or 300 mg/kg of uniformly labeled $^{13}$C$_5$-glutamine (Cambridge Isotope Laboratories, Cat # CLM-1822-H) and retinas were harvested 45 min later and snap-frozen as described above. Metabolites were extracted as described above and data collected according to previously published protocols (*Yuan et al., 2019*).

## SUrface SEnsing of translation (SUnSET) method

In vivo protein synthesis in whole retina from P14 WT and cKO mice was measured using the SUnSET protocol as previously described (*Fort et al., 2022*). Briefly, a stock solution of 40 mg/mL puromycin hydrochloride (Millipore-Sigma, Cat # P7255) was prepared in sterile 0.9% sodium chloride and injected intraperitoneally into mice at a final concentration of 200 mg/kg body weight. Mice were sacrificed after 30 min and fresh retinas were harvested and lysed in RIPA buffer (Thermo Fisher Scientific, Cat # 89900) with protease and phosphatase inhibitors (Cell Signaling Technology, Cat # 5872).

Protein quantitation was conducted with the BCA assay as described above and 10 µg of protein was analyzed by western blot analysis using an anti-puromycin antibody.

## BaroFuse

Oxygen consumption rate (OCR) was determine as previously described (*Kamat et al., 2023*). Briefly, a BaroFuse (Entox Sciences, Mercer Island, WA) was used to determine the OCR of freshly isolated retinas. Single, whole retinas were dissected into Hank's Balanced salt solution supplemented with 0.1 g/ 100 mL BSA (HBSS, Cytiva, Marlborough, MA, USA, Cat # SH30031FS). Perifusion media consisted of commercial Krebs-Ringer Bicarbonate buffer (KRB, Thermo Fisher Scientific, Cat # J67795.K2) supplemented with 0.1 g /100 mL fatty-acid free BSA (Millipore-Sigma, Cat # A9647) and 4.4 mM glucose, for a final concentration of 5.5 mM glucose (Millipore-Sigma, Cat # G8270). The oxygen and $CO_2$ concentration of the perifusion media is maintained by saturating the solution in a 21% oxygen, 5% $CO_2$ atmosphere, and the temperature was maintained at 37 °C throughout the experiment. At various times throughout the experiment Oligomycin-A (Cayman Chemical, Ann Arbor, MI, USA, Cat # 11342), FCCP (Trifluoromethoxy carbonylcyanide phenylhydrazone, Cayman Chemical, Cat# 15218), and KCN (Potassium Cyanide, Thermo Fisher, Cat # 012136) were added to the perifusion media through an injection port. A chamber without tissue was used as a negative control.

## NAD$^+$/NADH and NADP$^+$/NADPH measurements

The NAD$^+$/NADH and NADP$^+$/NADPH measurements were conducted using the NAD$^+$/NADH-Glo Assay (Promega, Cat # G9071) and NADP$^+$/NADPH-Glo Assay (Promega, Cat # G9081), respectively, following manufacturer's instructions in whole retina from P14 WT and cKO mice. Briefly, two fresh retina per mouse were harvested for each sample in 150 µL of PBS/Bicarbonate/0.5%dodecyltrime-thylammonimum bromide (DTAB) buffer. DTAB is used in the buffer to preserve dinucleotide stability. The samples were sonicated (20% strength and 1 s/10 iterations) to create a uniform suspension and diluted with 150 µL of PBS/Bicarbonate/0.5% DTAB buffer (lysate). For measuring NAD$^+$/NADP$^+$, 150 µL of lysate was mixed with 75 µL of 0.4 N HCl (Thermo Fisher Scientific, Cat # A144) and heated to 60 °C for 15 min. The lysate mixture was cooled to room temperature for 10 min and then neutralized by adding 150 µL 0.5 M Trizma base (Millipore-Sigma, Cat # T1503). For measuring NADH/NADPH, the remaining 150 µL of lysate was heated to 60 °C for 15 min, cooled to room temperature for 10 min and 150 µL 0.5 M Trizma base was added to the sample. Following lysate preparation, 50 µL of lysate was incubated at room temperature for 30 min with 50 µL of either the NAD+/NADP+-Glo or NADH/NAPDH-Glo detection reagent in a 96-well white walled tissue culture plate (Thermo Fisher Scientific, Cat # 3610). Luminescence was recorded using the Omega plate reader (BMG Labtech) and data are reported as a ratio of NAD$^+$/NADH or NADP$^+$/NADPH of N=3 animals in triplicate technical measurements per animal sample.

## Statistical analysis

All data is presented as mean ± SEM. The significance of the difference between means was determined using a two-tailed student's *t*-test or one-way ANOVA in Excel or Prism 9.0. Results with a p-value ≤0.05 were considered significant.

## Acknowledgements

Funding for this research was supported by a NEI K08EY031757, a RPB Unrestricted Grant, the Retina Society Research and Education grant, and the Global Ophthalmology Awards Program (GOAP), a Bayer-sponsored initiative committed to supporting ophthalmic research across the world. This work utilized the Vision Research Core funded by P30 EY007003 from the National Eye Institute. CAL was supported by the following grants: R37CA237421, R01CA248160, R01CA244931, and P01HL149633. CGB was supported by R01EY029675. We thank Dr. Stephen Rayport in the Department of Psychiatry, Columbia University, New York, NY, for providing the *Gls*$^{flox/flox}$ mice. We kindly thank Dr. Stephen Tsang in the Department of Ophthalmology, Columbia University, New York, NY, for the photoreceptor-specific, tamoxifen inducible mouse model (*Pde6g*$^{Cre:ERT2}$).

# Additional information

## Competing interests

Cagri Besirli: Owns Johnson & Johnson stock and has equity interest in Ocutheia and iRenix Medical. Costas A Lyssiotis: Has consulted for Astellas Pharmaceuticals, Odyssey Therapeutics, Third Rock Ventures, and T-Knife Therapeutics; is an inventor on patents pertaining to Kras regulated metabolic pathways, redox control pathways in pancreatic cancer, and targeting the GOT1-ME1 pathway as a therapeutic approach (US Patent No: 2015126580-A1, 05/07/2015; US Patent No: 20190136238, 05/09/2019; International Patent No: WO2013177426-A2, 04/23/2015). Thomas J Wubben: Has equity interest in Ocutheia. The other authors declare that no competing interests exist.

## Funding

| Funder | Grant reference number | Author |
|---|---|---|
| National Eye Institute | K08EY031757 | Thomas J Wubben |
| Research to Prevent Blindness | | Thomas J Wubben |
| Retina Society Research and Education | | Thomas J Wubben |
| Global Ophthalmology Awards Program | | Thomas J Wubben |
| National Eye Institute | P30 EY007003 | Moloy T Goswami<br>Eric Weh<br>Shubha Subramanya<br>Katherine M Weh<br>Hima Bindu Durumutla<br>Heather Hager<br>Nicholas Miller<br>Sraboni Chaudhury<br>Cagri Besirli<br>Thomas J Wubben |
| National Cancer Institute | R37CA237421 | Costas A Lyssiotis |
| National Cancer Institute | R01CA248160 | Costas A Lyssiotis |
| National Cancer Institute | R01CA244931 | Costas A Lyssiotis |
| National Heart, Lung, and Blood Institute | P01HL149633 | Costas A Lyssiotis |
| National Eye Institute | R01EY029675 | Cagri Besirli |

The funders had no role in study design, data collection and interpretation, or the decision to submit the work for publication.

## Author contributions

Moloy T Goswami, Conceptualization, Formal analysis, Investigation, Writing – review and editing; Eric Weh, Conceptualization, Formal analysis, Supervision, Investigation, Visualization, Writing – original draft, Writing – review and editing; Shubha Subramanya, Formal analysis, Investigation, Visualization, Writing – review and editing; Katherine M Weh, Visualization, Writing – review and editing; Hima Bindu Durumutla, Heather Hager, Nicholas Miller, Sraboni Chaudhury, Investigation; Anthony Andren, Peter Sajjakulnukit, Li Zhang, Resources; Cagri Besirli, Supervision, Writing – review and editing; Costas A Lyssiotis, Conceptualization, Resources, Supervision, Methodology; Thomas J Wubben, Conceptualization, Formal analysis, Supervision, Funding acquisition, Visualization, Methodology, Writing – original draft, Writing – review and editing

## Author ORCIDs

Shubha Subramanya ⬤ https://orcid.org/0009-0002-6976-575X
Katherine M Weh ⬤ https://orcid.org/0000-0002-7745-5391
Costas A Lyssiotis ⬤ https://orcid.org/0000-0001-9309-6141
Thomas J Wubben ⬤ https://orcid.org/0000-0002-5183-3817

### Ethics

All animals were treated in accordance with the Association for Research in Vision and Ophthalmology Statement for the Use of Animals in Ophthalmic and Vision Research. The protocol was approved by the University Committee on Use and Care of Animals of the University of Michigan (PRO00011133).

Reviewer #1 (Public review): https://doi.org/10.7554/eLife.100747.3.sa1
Reviewer #2 (Public review): https://doi.org/10.7554/eLife.100747.3.sa2
Reviewer #3 (Public review): https://doi.org/10.7554/eLife.100747.3.sa3
Author response https://doi.org/10.7554/eLife.100747.3.sa4

## Additional files

### Supplementary files

Supplementary file 1. Primers utilized in qRT-PCR experiments to measure gene expression.

Supplementary file 2. List of metabolites and their parameters detected by LC-MS/MS.

MDAR checklist

### Data availability

All data generated or analyzed during this study are included in the manuscript and supporting files; source data files have been provided for Figures 1, 3, 4, and 6.

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
