## [Editor Report · eLife Assessment]

Goswami and colleagues used rod-specific Gls1 (the gene encoding glutaminase 1) knockout mice to investigate the role of GLS1 in photoreceptor health when GLS1 was deleted from developing or adult photoreceptor cells. This study is **fundamental** as it shows the critical role of glutamine catabolism in photoreceptor cell health using in vivo model systems. The evidence supporting the authors' claims is **compelling**. The studies add new insight into how specific metabolites support vision.

---

## [Referee Report · Reviewer #1 (Public review)]

Summary:

The authors show for the first time that deleting GLS from rod photoreceptors results in the rapid death of these cells. The death of photoreceptor cells could result from loss of synaptic activity because of a decrease in glutamate, as has been shown in neurons, changes in redox balance, or nutrient deprivation.

Strengths:

The strength of this manuscript is that the author shows a similar phenotype in the mice when Gls was knocked out early in rod development or the adult rod. They showed that rapid cell death is through apoptosis, and there is an increase in the expression of genes responsive to oxidative stress.

Comments on revisions:

The authors addressed all of my concerns in their responses to reviewers.

---

## [Referee Report · Reviewer #2 (Public review)]

Summary:

Photoreceptor neurons are crucial for vision, and discovering pathways necessary for photoreceptor health and survival can open new avenues for therapeutics. Studies have shown that metabolic dysfunction can cause photoreceptor degeneration and vision loss, but the metabolic pathways maintaining photoreceptor health are not well understood. This is a fundamental study that shows that glutamine catabolism is critical for photoreceptor cell health using in vivo model systems.

Strengths:

The data are compelling, and the consideration of potential confounding factors (such as glutaminase 2 expression) and additional experiments to examine the synaptic connectivity and inner retina added strength to this work. The authors were also careful not to overstate their claims, but to provide solid conclusions that fit the results and data provided in their study. The findings linking asparagine supplementation and the inhibition of the integrated stress response to glutamine catabolism within the rod photoreceptor cell are intriguing and innovative. Overall, the authors provide convincing data to highlight that photoreceptors utilize various fuel sources to meet their metabolic needs, and that glutamine is critical to these cells for their biomass, redox balance, function and survival.

---

## [Referee Report · Reviewer #3 (Public review)]

Summary:

The authors explored the role of GLS, a glutaminase, which is an enzyme catalyzes the conversion of glutamine to glutamate, in rod photoreceptor function and survival. The loss of GLS was found to cause rapid autonomous death of rod photoreceptors.

Strengths:

Interesting and novel phenotype. Two types of cre-lines were rigorously used to knockout Gls gene in rods. Both of the conditional knockouts led to a similar phenotype, i.e. rod death. Histology and ERG were carefully done to characterize the loss of rods over specific ages. Necessary metabolomic study was performed and appreciated. Some rescue experiments were performed, and revealed possible mechanism.

Weaknesses:

No major weaknesses. Mechanism of GLS-loss induced rod death could be followed up in the future, and same for GLS's role in cones. Authors have addressed all minor points raised by this reviewer.

---

## [Author Response]

The following is the authors’ response to the original reviews

**Public Reviews:**

**Reviewer #1 (Public Review):**
Summary:The authors show for the first time that deleting GLS from rod photoreceptors results in the rapid death of these cells. The death of photoreceptor cells could result from loss of synaptic activity because of a decrease in glutamate, as has been shown in neurons, changes in redox balance, or nutrient deprivation.Strengths:The strength of this manuscript is that the author shows a similar phenotype in the mice when Gls was knocked out early in rod development or the adult rod. They showed that rapid cell death is through apoptosis, and there is an increase in the expression of genes responsive to oxidative stress.

We thank the reviewer for their time reviewing the manuscript and their comments regarding the potential mechanism(s) by which rod photoreceptors rapidly degenerate upon knockout of GLS.

Weaknesses:In this manuscript, the authors show a "metabolic dependency of photoreceptors on glutamine catabolism in vivo". However, there is a potential bias in their thinking that glutamine metabolism in rods is similar to cancer cells where it feeds into the TCA cycle. They should consider that as in neurons, GLS1 activity provides glutamate for synaptic transmission. The modest rescue shown by providing α-ketoglutarate in the drinking water suggests that glutamine isn't a key metabolic substrate for rods when glucose is plentiful. The ERG studies performed on the iCre-Glsflox/flox mice showed a large decrease in the scotopic b wave at saturating flashes which could indicate a decrease in glutamate at the rod synapse as stated by the authors. While EM micrographs of wt and iCre-Glsflox/flox mice were shown for the outer retina at p14, the synapse of the rods needs to be examined by EM.

We agree with the reviewer that in the presence of sufficient glucose, it appears a lack of GLS-driven glutamine (Gln) catabolism does not drastically alter the levels of TCA cycle metabolites or mitochondrial function as we demonstrated in Figure 4, and supplementation with alpha-ketoglutarate improved outer nuclear layer thickness by only a small amount as observed in Figure 5e. Hence, as we stated in the Results and Discussion, at least in the mouse where *Gls* is selectively deleted from rod photoreceptors by crossing *Glsfl/fl* mice with *Rho-Cre* mice (*Glsfl/fl; Rho-Cre*^+^*,* cKO), Gln’s role in supporting the TCA cycle is not the major mechanism by which rod photoreceptors utilize Gln to suppress apoptosis.

With regards to GLS-driven Gln catabolism providing glutamate (Glu) for synaptic transmission, we again agree with the reviewer that Glu is an important excitatory neurotransmitter, but it is also a key metabolite necessary for the synthesis of glutathione, amino acids, and proteins. As noted and discussed at length in the manuscript, a lack of GLS-driven Gln catabolism in rod photoreceptors leads to reduced levels of oxidized glutathione (Figure 4D) possibly signaling an overall reduction in the biosynthesis of glutathione as Glu is directly and indirectly responsible for its synthesis. Furthermore, Gln and GLS-derived Glu play a central role in the biosynthesis of several nonessential amino acids and proteins. To this end, we see a reduction in the level of Glu, which is the product of the GLS reaction and further confirms the loss of GLS function. We also noted a significant decrease in aspartate (Asp), which can be constructed from the carbons and nitrogens of Gln as discussed at length in the manuscript (Figure 6A). Finally, we noted a significant decrease in global protein synthesis in the cKO retina as compared to the wild-type animal as well (Figure 6E). Therefore, the data suggest that GLS-driven Gln catabolism is critical for amino acid metabolism and protein synthesis and to some degree redox balance; although, the small but statistically significant changes in oxidized glutathione, NADP/NADPH, and redox gene expression may not fully account for the rapid and complete photoreceptor degeneration observed. Future studies are necessary to shed light on the role of redox imbalance in this novel transgenic mouse model.

Glu also plays a role in synaptic transmission, and we considered this scenario as described in Figure 1 – figure supplement 5. Here, the synaptic connectivity between photoreceptors and the inner retina did not demonstrate significant differences in the labeling of photoreceptor synaptic membranes in the outer plexiform layer nor alterations in the labeling of a key protein (Bassoon) in ribbon synapses. These data suggest that the synaptic connectivity between photoreceptors and second-order neurons was unaltered at P14 in the cKO retina, which is the time just prior to rapid photoreceptor degeneration when Glu was shown to be decreased (Figure 6A).

With regards to the ERG changes noted in Figure 2, we agree with the reviewer that a large decrease was noted in the scotopic b-wave at P21 and P42 in the cKO. We also agree, that to obtain greater insight into these ERG changes, the ribbon synapse in EM images can be examined. The EM images shown in Figure 1 – figure supplement 4 are from P21, which coincide with the age at which the ERG changes were first noted and when significant photoreceptor degeneration has already occurred. These images were utilized to assess the ribbon synapse for the revised version of the manuscript. As now shown in Figure 1 – figure supplement 4D, ribbon synapses are intact in WT animals as denoted by the yellow boxes. Similarly, the ribbons (yellow arrows) appear structurally intact in the photoreceptors that remain in the P21 cKO retina. These results are in accordance with the lack of significant differences in the labeling of photoreceptor synaptic membranes in the outer plexiform layer as well as the lack of alterations in the labeling of a key protein (Bassoon) in ribbon synapses (Figure 1-figure supplement 5A and B). While we cannot fully rule out that the decrease in glutamate is altering synaptic transmission, our structural data suggests the synapses remain intact. These data have been added to the revised manuscript.

However, an even larger reduction in the scotopic a-wave was noted at these ages as well. In animal models that disrupt photoreceptor synaptic function (Dick et al. Neuron. 2003; Johnson et al. J Neuroscience. 2007; Haeseleer et al. Nature Neuroscience. 2004; Chang et al. Vis Neurosci. 2006), a more negative ERG pattern is typically observed with the b-wave altered to a much larger degree than the a-wave. Additionally, in these models that disrupt photoreceptor synaptic transmission, the overall structure of the retina with respect to thickness is maintained (Dick et al. Neuron. 2003) or noted to have modest changes in the outer plexiform layer within the first two months of age with the outer nuclear layer not significantly altered until 8-10 months of age (Haeseleer et al. Nature Neuroscience. 2004). In contrast, a rapid decline in the outer nuclear layer thickness was observed in the cKO retina after P14 likely contributing to the ERG changes noted in Figure 2. Also, Gln is catabolized to Glu primarily by GLS as suggested by the approximately 50% reduction in Glu levels in the cKO retina (Figure 6A), but other enzymes are also capable of catabolizing Gln to Glu, so Glu levels in the rod photoreceptors are unlikely to be zero. Coupling this with the fact that rods are equipped with a self-sufficient Glu recollecting system at their synaptic terminals (Hasegawa et al. Neuron. 2006; Winkler et al. Vis Neurosci. 1999) and that GLS activity is at least two-fold higher in the photoreceptor inner segments, which support energy production and metabolism, than any other layer in the retina (Ross et al. Brain Res. 1987) suggests that altered synaptic transmission secondary to reduced levels of Glu likely does not account in full for the rapid and robust photoreceptor degeneration observed in the cKO retina.

The authors note that the outer segments are shorter but they do not address whether there is a decrease in the number of cones.

We have adjusted Figure 2E by removing the GLS staining to better highlight the secondary degeneration of cone outer segments, the main point of the Figure, as we had already shown that GLS was cleanly knocked out of rod photoreceptors in Figure 1. Furthermore, qualitatively the number of cones appears the same at P14, P21, and P42 between the WT and cKO, which is consistent with other retinal degeneration models, like *rd1* and *rd10*, where cones do not begin to die until all the rods have degenerated (Xue et al. eLife. 2021).

Rod-specific Gls ko mice with an inducible promoter were generated by crossing the Pde6g-CreERT2 and homozygous for either the WT or floxed Gls allele (IND-cKO). In Figure 3 the authors document that by western blots and antibody labeling the GLS1 expression is lost in the IND-cKO 10 days post tamoxifen. OCT images show a decrease in the thickness of the outer nuclear layer between 17 and 38 days post-TAM. Ergs should be performed on the animals at 10 and 30 days post TAM, before and after major structural changes in rod photoreceptor cells, to determine if changes in light-stimulated responses are observed. These studies could help to parse out the cause of photoreceptor cell death.

We agree with the reviewer that the IND-cKO is a useful tool to help parse out the cause of photoreceptor cell death in this model as well as shed light on the role of GLS-driven Gln catabolism in photoreceptor synaptic transmission as discussed at length above. Hence, ERG analyses were performed 10 days post TAM, before major structural changes in the ONL are observed. Interestingly, ERG demonstrated statistically significant reductions in the IND-cKO scotopic a- and b-waves as compared to the WT 10 days post TAM. Similarly, photopic ERG demonstrated statistically significant decreases in the b-wave of the IND-cKO retina. These data suggest that GLS-driven Gln catabolism plays a significant role not only in rod photoreceptor survival but their function as well. This data has been added to Figure 3H-I and discussed in the corresponding manuscript text.

To this end, as discussed below and added to Figure 6 – figure supplement 1, amino acid levels, including glutamate (Glu), are already reduced 10 days post TAM. Reductions in the level of Glu may impact synaptic transmission and as a result, the scotopic b-wave. However, as noted above, altered synaptic transmission secondary to reduced levels of Glu likely does not account in full for the rapid and robust photoreceptor degeneration observed in the cKO retina as the b-wave to a-wave ratio is not significantly altered in the IND-cKO retina as compared to the WT retina, suggesting GLS-driven Gln catabolism is impairing both to a similar degree.

Additionally, *Pde6g* is expressed by rods to a significant degree but also by cones (GSE63473, scRNAseq data). Therefore, the IND-cKO mouse likely knocks out GLS from both rods and cones, which is in accordance with the immunofluorescence image in Figure 3B where GLS is not observed in rod or cone inner segments unlike in Figure 1B where GLS remains in cones. Hence, the reduction in photopic b-wave may be demonstrating that GLS-driven Gln catabolism in cones impairs synaptic transmission. As noted in our reply to reviewer #3’s comments, we have generated mice lacking GLS in cone photoreceptors specifically and are currently elucidating the role of GLS in cone photoreceptor metabolism, function, and survival. These results will be published in a separate manuscript.

The studies in Figure 4 were all performed on iCre-Glsflox/flox and control mice at p14, why weren't the IND-cKO mice used for these studies since the findings would not be confounded by development?

To gain further insight into the role of GLS-driven Gln catabolism in the maintenance of rod photoreceptors as compared to their development/maturation, we conducted a targeted metabolomic analysis on IND-cKO and WT retinas 10 days post TAM. For the purpose of this manuscript, we have included data regarding changes in amino acid levels in Figure 6 – figure supplement 1. Specifically, levels of glutamate, aspartate and asparagine are all significantly decreased in the IND-cKO retina prior to PR degeneration, which demonstrates that similar to the GLS cKO mouse (i.e. iCre-Gls flox/flox), GLS-driven Gln catabolism is critical for amino acid biosynthesis in mature rod PRs as well.

In all rescue studies, the endpoint was an ONL thickness, which only addressed rod cell death. The authors should also determine whether there are small improvements in the ERG, which would distinguish the role of GLS in preventing oxidative stress.

Optical coherence tomography (OCT) provides a sensitive in vivo method to detect small changes in retinal thickness without potential artifacts incurred through histological processing. Considering the *Gls* cKO retina demonstrates significant and rapid photoreceptor degeneration, we wanted to assess pathways that may be critical to photoreceptor survival downstream of GLS-driven Gln catabolism using rescue experiments with pharmacologic treatment or metabolite supplementation. That said, disruption of GLS-driven Gln catabolism may also significantly alter rod photoreceptor function beyond that which is secondary to photoreceptor cell death as we have demonstrated in the IND-cKO animal for the revised version of this manuscript and discussed in a response above. Therefore, the IND-cKO model provides a unique tool to assess the impact of rescue studies on photoreceptor function as the functional changes occur prior to significant degeneration. Also, unlike the GLS cKO mouse (i.e. iCre-Gls flox/flox) where photoreceptor degeneration starts very early, impairing our ability to capture reliable and robust ERG measurements, the IND-cKO mice are older at the time of functional changes allowing for robust ERG measurements. While the rate of photoreceptor degeneration in both mouse models is similar and the levels of key amino acids are altered similarly in both models, the mechanisms of cell death in developing/maturing photoreceptors may be different than that in mature photoreceptors. Hence, before we can assess if similar rescue experiments impact photoreceptor function via ERG in the IND-cKO mouse, we need to thoroughly examine how these photoreceptors are dying. These experiments and results will be published in a separate manuscript in the future.

**Reviewer #2 (Public Review):**
Summary:Photoreceptor neurons are crucial for vision, and discovering pathways necessary for photoreceptor health and survival can open new avenues for therapeutics. Studies have shown that metabolic dysfunction can cause photoreceptor degeneration and vision loss, but the metabolic pathways maintaining photoreceptor health are not well understood. This is a fundamental study that shows that glutamine catabolism is critical for photoreceptor cell health using in vivo model systems.Strengths:The data are compelling, and the consideration of potential confounding factors (such as glutaminase 2 expression) and additional experiments to examine the synaptic connectivity and inner retina added strength to this work. The authors were also careful not to overstate their claims, but to provide solid conclusions that fit the results and data provided in their study. The findings linking asparagine supplementation and the inhibition of the integrated stress response to glutamine catabolism within the rod photoreceptor cell are intriguing and innovative. Overall, the authors provide convincing data to highlight that photoreceptors utilize various fuel sources to meet their metabolic needs, and that glutamine is critical to these cells for their biomass, redox balance, function, and survival.

We greatly appreciate the reviewer’s thoughtful comments and time spent reviewing this manuscript.

Weaknesses:Recent studies have explored the metabolic "crosstalk" that exists within the mammalian retina, where metabolites are transferred between the various retinal cells and the retinal pigment epithelium. It would be of interest to test whether the conditional knockout mice have changes in metabolism (via qPCR such as shown in Figure 4 - Supplemental Figure 1) within the retinal pigment epithelium that may be contributing to the authors' findings in the neural retina. Additionally, the authors have very compelling data to show that inhibition of eIF2a or supplementation with asparagine can delay photoreceptor death via OCT measurements in their conditional knockout mouse model (Figure 6G, H). However, does inhibition of eIF2a or asparagine adversely impact the WT retina? It would also be impactful to know whether this has a prolonged effect, or if it is short-term, as this would provide strength to potential therapeutic targeting of these pathways to maintain photoreceptor health.

We agree with the reviewer that metabolic communication in the outer retina is crucial to the function and survival of both photoreceptors and RPE. Therefore, we have performed qRT-PCR on eyecups from cKO and WT mice at P14, prior to photoreceptor degeneration. These data, now included in Figure 4 – figure supplement 2, show no significant changes in genes related to glycolysis, pyruvate metabolism and the TCA cycle in eyecups from cKO mice compared to WT mice at P14. The only exception is a significant decrease in *Pdk4* in cKO mouse eyecups compared to WT, which was not observed in retina samples.

Additionally, we have added data demonstrating that systemic treatment with ISRIB does not adversely impact the anatomy of the wild-type retina. Specifically, we performed OCT after 21 days of ISRIB treatment via intraperitoneal delivery in WT mice and show that total retinal, ONL and inner segment/outer segment thickness is unchanged compared to vehicle. These data are now included in Figure 6 – figure supplement 2A. We have also included data to suggest that the effect of ISRIB extends beyond P21 in the cKO mouse. This data, presented in Figure 6 – figure supplement 2B, shows that at P28, ISRIB continues to statistically significantly increase ONL thickness compared to vehicle in cKO animals.

**Reviewer #3 (Public Review):**
Summary:The authors explored the role of GLS, a glutaminase, which is an enzyme that catalyzes the conversion of glutamine to glutamate, in rod photoreceptor function and survival. The loss of GLS was found to cause rapid autonomous death of rod photoreceptors.Strengths:Interesting and novel phenotype. Two types of cre-lines were rigorously used to knockout the Gls gene in rods. Both of the conditional knockouts led to a similar phenotype, i.e. rod death. Histology and ERG were carefully done to characterize the loss of rods over specific ages. A necessary metabolomic study was performed and appreciated. Some rescue experiments were performed and revealed possible mechanisms.

We thank the reviewer for their comments and appreciation of the methods utilized herein to address the role of GLS-driven Gln catabolism in rod photoreceptors.

Weaknesses:No major weaknesses were identified. The mechanism of GLS-loss-induced rod death seems not fully elucidated by this study but could be followed up in the future, and the same for GLS's role in cones.

We agree with the reviewer that the downstream metabolic and molecular mechanisms by which Gln catabolism impacts rod photoreceptor health are not fully elucidated. Defining these mechanisms will advance our understanding of photoreceptor metabolism and identify therapeutic targets promoting photoreceptor resistance to stress. Future studies are underway to uncover these mechanisms. Additionally, while outside the scope of the current manuscript, we have generated mice lacking GLS in cone photoreceptors specifically and are currently elucidating the role of GLS in cone photoreceptor metabolism, function, and survival. These results will be published in a separate manuscript.

**Reviewer #1 (Recommendations For The Authors):**
(1) The results could start at line 135, but the first paragraph isn't necessary. The data is published and could be referred to in the introduction.

We appreciate the reviewer’s suggestion to shorten the beginning of the Results section; however, we believe the supplementary data, which is described in these lines, confirms the scRNAseq gene expression data, while adding GLS expression and localization data within the retina. The scRNAseq data and its publication was noted in the introduction, so we removed the sentence in line 117-119 that restates these results to shorten this section. We also reduced redundancy by removing an introductory sentence to the second Results paragraph.

(2) "However, like other metabolically-demanding cells, recent work has demonstrated that PRs have the flexibility to utilize fuel sources beyond glucose to meet their metabolic needs (Adler et al., 2014; Du, Cleghorn, Contreras, Linton, et al., 2013; Grenell et al., 2019; Joyal et al., 2016; Xu et al., 2020)." The paper by Daniele et al. demonstrated that glucose is essential for maintaining the viability of rod photoreceptor cells.

We thank the reviewer for highlighting published literature, which we apologetically overlooked. The reference for Daniele et al. has now been included.

(3) "Single-cell RNA sequencing data has demonstrated that Gls is expressed throughout the human and mouse retina and much greater than Gls2 (Voigt et al., 2020). The authors should indicate the specific databases searched in Spectacle.

We appreciate the reviewer’s attention to detail and have now included the references in the Introduction for GSE63473 from Macosko et al. and GSE142449 from Voigt et al., which were the databases we used in Spectacle to assess *Gls* levels in the mouse and human retina, respectively.

References:

(1) Macosko EZ, Basu A, Satija R, Nemesh J, Shekhar K, Goldman M, Tirosh I, Bialas AR, Kamitaki N, Martersteck EM, Trombetta JJ, Weitz DA, Sanes JR, Shalek AK, Regev A, McCarroll SA. Highly Parallel Genome-wide Expression Profiling of Individual Cells Using Nanoliter Droplets. Cell. 2015 May 21;161(5):1202-1214. doi: 10.1016/j.cell.2015.05.002. PMID: 26000488; PMCID: PMC4481139.

(2) Voigt AP, Binkley E, Flamme-Wiese MJ, Zeng S, DeLuca AP, Scheetz TE, Tucker BA, Mullins RF, Stone EM. Single-Cell RNA Sequencing in Human Retinal Degeneration Reveals Distinct Glial Cell Populations. Cells. 2020 Feb 13;9(2):438. doi: 10.3390/cells9020438. PMID: 32069977; PMCID: PMC7072666.

(4) The immunolabeling in Figure 2 looks like the images are overexposed, and the Gls antibody is labeling the outer segment, not just the inner segment of photoreceptors.

We thank the reviewer for their comments regarding our immunofluorescence data. There was background staining of the outer segment in both the WT and cKO retina with decreased GLS staining in the inner segment of the cKO rod photoreceptors at P14 demonstrating loss of GLS in rod photoreceptors similar to Figure 1B. For Figure 2E, we have provided adjusted images with PNA staining only that better represent the secondary cone degeneration that occurs in the rod photoreceptor-specific *Gls* cKO, which is the take home point of Figure 2E.

(5) The authors could use a glutamate antibody to compare it to Gls KO mice as done in Davanger, S., Ottersen, O.P. and Storm-Mathisen, J. (1991), Glutamate, GABA, and glycine in the human retina: An immunocytochemical investigation. J. Comp. Neurol., 311: 483-494. https://doi.org/10.1002/cne.903110404

We appreciate the reviewer’s suggestion to assess glutamate levels in the wild-type and *Gls* KO retina via antibody labeling. Our targeted metabolomics studies in Figure 6A provide quantitative evidence that glutamate, the product of the GLS-catalyzed reaction, is decreased as one would expect in that *Gls* KO retina. The antibody would add to these data by providing the localization of glutamate in the retina. With a rod photoreceptor-specific genetic KO, we would expect glutamate levels to be decreased in these cells. The antibody may also show that glutamate is not only decreased in the rod photoreceptor inner segment, where GLS predominates, but also in the synaptic terminal in accordance with the reviewer’s concerns regarding the impact of GLS KO on synaptic transmission. We have addressed this concern at length above, adding TEM images of the ribbon synapses in the GLS KO retina, and ERG analyses from the IND-cKO animals prior to significant degeneration. In the end, we agree with the reviewer that reduced Glu levels in the GLS cKO retina may impact synaptic transmission to a degree, but the synapses remain intact based on immunofluorescence and TEM analyses and a negative ERG pattern is not observed in the GLS cKO (i.e. iCre-Gls flox/flox) or IND-cKO mouse. As noted above, the structure of the retina in models that disrupt photoreceptor synaptic transmission is maintained (Dick et al. Neuron. 2003) or noted to have modest changes within the first two months of age with the outer nuclear layer not significantly altered until 8-10 months of age (Haeseleer et al. Nature Neuroscience. 2004). So, the impact of the reduced Glu levels on synaptic transmission in the GLS KO retina are unlikely to account in full for the rapid and profound photoreceptor degeneration observed. That said, the IND-cKO mouse, which allows us to assess photoreceptor function prior to significant degeneration unlike the GLS cKO mouse (i.e. iCre-Gls flox/flox), demonstrates GLS-driven Gln catabolism plays a significant role in photoreceptor function but still does not demonstrate a negative ERG pattern. Therefore, assessing Glu localization in this mouse model 10 days post TAM will be informative as to how GLS-driven Gln catabolism impacts photoreceptor function prior to degeneration. The IND-cKO mouse model is currently being extensively characterized for future publication.

**Reviewer #2 (Recommendations For The Authors):**
Main Concerns:(1) The authors checked for Gls2 compensation at P14 in the mouse retina. However, this data would be more compelling with an additional timepoint, particularly at P21 which is used in many of their figures throughout the study.

We thank the reviewer for their suggestion. Figure 1-figure supplement 1D demonstrates no change in *Gls2* gene expression at P14 between the WT and cKO retina. With regards to the reviewer’s concern, in Figure 1-figure supplement 1E of the original submission, we demonstrate that the expression of GLS2 is not increased in the cKO retina at P21 via immunofluorescence.

(2) Recent studies have explored the metabolic "crosstalk" that exists within the mammalian retina, where metabolites are transferred between the various retinal cells and the retinal pigment epithelium. It would be compelling to see whether the cKO mice have changes in metabolism (via qPCR such as shown in Supplementary Figure 1 for Figure 4) within the RPE that may be contributing to their findings in the neural retina. Additionally, mention of this crosstalk and how it may impact their results should be added to the discussion.

We appreciate the reviewer’s concern for metabolism changes in the RPE of *Gls* cKO mice. In agreement with reviewer 2, we performed qRT-PCR on eyecups from cKO and WT mice at P14, prior to photoreceptor degeneration. These data, now included in Figure 4 – figure supplement 2, show no significant changes in genes related to glycolysis, pyruvate metabolism and the TCA cycle in eyecups from cKO mice compared to WT mice at P14. The only exception is a significant decrease in *Pdk4* in cKO mouse eyecups compared to WT, which was not observed in retina samples.

(3) The authors use a tamoxifen-inducible cKO model to support their findings in developed rods. However, in Figure 3A it appears that this model has a greater reduction in GLS compared to the Rho-cre mouse model. Can the authors discuss this? Is this cre more efficient at targeting rods or is it leaky and may have affected other retinal cells?

We thank the reviewer for pointing out this interesting result associated with using the *Pde6g*-Cre-ERT2 mouse line. *Pde6g* is expressed by rods to a significant degree but also by cones (GSE63473, scRNAseq data). Therefore, the IND-cKO mouse likely knocks out GLS from both rods and cones upon the TAM induction. To this end, the immunofluorescence image in Figure 3B shows GLS is knocked out in both rod or cone inner segments unlike in Figure 1B where GLS remains in cones when using the rod photoreceptor-specific, *Glsfl/fl Rho-Cre*^+^ mouse. As such, as the astute reviewer noted, the fact that Western blot demonstrates greater reduction in GLS protein content fits with the protein being knocked out of both rods and cones. We have added this note about the mouse model in the corresponding text.

(4) The authors have very compelling data to show that inhibition of eIF2a can delay photoreceptor death via OCT measurements in their cKO mouse model (Figure 6G). However, does ISRIB adversely impact the WT retina? WT vehicle and ISRIB should be shown. It would also be compelling to know whether this has a prolonged effect, or if it is short-term (i.e. would the effect still be present at P42)?

We appreciate the reviewer’s comments regarding antagonizing the effects of p-eIF2a to prolong photoreceptor survival in the *Gls* cKO retina. As described above, we have data demonstrating systemic treatment with ISRIB does not adversely impact the anatomy of the wild-type retina (Figure 6-figure supplement 2A). Specifically, we treated WT animals with daily intraperitoneal ISRIB starting at P5 and performed OCT at P21 to show that total retinal, ONL and the inner segment/outer segment thickness is unchanged compared to vehicle-treated WT animals. Additionally, we have included data demonstrating the photoreceptor neuroprotective effect of ISRIB treatment in the *Gls* cKO mouse extends beyond P21 in the cKO mouse (Figure 6-figure supplement 2B).

(5) For Figure 6H, same as point #4.

While we have not specifically assessed potential retinal toxicity secondary to systemic Asn supplementation, oral Asn supplementation (up to 100mg/kg/day) was provided to patients for 24 months and found to be well-tolerated (PMID:31123592). Allometric scaling of this dose to the mouse would yield a mouse dose of 1234 mg/kg/day, which is much greater than the 200mg/kg/day dose provided here (PMID: 27057123). Additionally, a 90-day toxicity study of Asn in rats demonstrated a no observed adverse effect level of 1.62g/kg bodyweight/day in males and 1.73g/kg bodyweight/day in females (PMID: 18508175). The lower dose in that study equates to a mouse dose of 3.2g/kg bodyweight/day, well above the mouse dose utilized in this report. As such, future studies should focus on a dose-response relationship with Asn supplementation, and as the reviewer suggested, determining the duration of effect with Asn supplementation.

(6) Some of the results section belongs in the introduction or discussion and can be moved.

We have addressed the reviewer’s concern by moving some of the results to the discussion and removing statements in the results that were either noted in the Introduction or conferred in the Discussion.

Minor Concerns:(1) Scale bar mentions in the figure legends use plural when only one is present, or in some cases are missing. A scale bar should be added to the OCT images if possible.

We appreciate the reviewer’s attention to detail, and information regarding scale bars has been updated in the figure legends.

(2) For Figures 1I and J, the sample size changes when J is a quantification of I. Please correct.

We have corrected the sample size to be consistent between Figures 1I and J.

(3) In Figure 1 - Figure Supplement 3 the P42 timepoint is not mentioned in the legend. Please correct.

We have now included the P42 timepoint in the legend for in Figure 1 – Figure Supplement 3 as well as the manuscript text.

(4) In Figure 1 - Figure Supplement 5 the wrong P value is mentioned in the legend. Please correct.

We have corrected the P value in the legend for Figure 1 – Figure Supplement 5.

(5) Can the authors double-check their ERG light intensity settings? They seem high. Please confirm if they are correct.

We appreciate the reviewer’s concern for ERG light intensity settings and have confirmed the settings used in the study were 32 cd*s/m^2^ and 100 cd*s/m^2^ for scotopic and photopic ERG recordings, respectively.

(6) The legend key in Figure 2A would be more helpful if the axis were present by the representative traces.

We thank the reviewer for the suggestion of adding axes to the ERG traces. Figure 2A has been updated to reflect this modification.

(7) Can the authors check that the error bars are present in Figure 5E?

We appreciate the reviewer’s concern for error bars in Figure 5E, which are included in the figure. The standard error in this experiment is so small that the symbols overlap with the error bars.

**Reviewer #3 (Recommendations For The Authors):**
Suggestions for improved or additional experiments, data, or analyses.(1) Figure 6: ISRIB seems to give the most dramatic rescue of cKO GLS in P21 rods. Does it completely prevent rod death? i.e. What's the ONL thickness of P21 WT control? What's the ISRIB rescue of an older cKO animal, say P35?

The ONL thickness of P21 WT control is on average 0.06 mm (Figure 1E), while the ONL thickness of the *Gls* cKO retina with ISRIB treatment at P21 is on average 0.044 mm. Therefore, rod death is not completely prevented with ISRIB but rather, rod photoreceptor survival is prolonged. As noted above, we have provided data to demonstrate that the photoreceptor neuroprotective effect of ISRIB lasts beyond P21 (Figure 6-figure supplement 2B).

(2) What's the mechanistic link between ISR and GLS beyond current speculation? Does GLS have other unknown functions beyond converting glutamine to glutamate? Any novel insights from GLS protein structure?

We thank the reviewer for this thoughtful question. It is certainly possible that GLS has other functions outside of its role in glutaminolysis. It is well known that other metabolic enzymes have moonlighting functions including hexokinase 2, which has been shown to be important in preventing intrinsic apoptosis through blocking the binding of pro-apoptotic proteins to the mitochondria. While not directly related to ISR, a single report suggests GLS functions non-canonically in Gln-deprived states, promoting mitochondrial fusion to suppress ROS production (PMID: 29934617). Investigating the moonlighting functions of metabolic enzymes is part of our ongoing research program and GLS is included in these studies.

(3) Just curious about GLS cKO in cones. Any similar phenotype?

We appreciate the reviewer’s curiosity regarding *Gls* cKO in cones and this study is currently ongoing with a poster presented at ARVO 2024 (Subramanya et al; Glutaminase-driven glutamine catabolism supports cone photoreceptor metabolism, function, and structure. Invest. Ophthalmol. Vis. Sci. 2024;65(7):193) and a manuscript in preparation. As discussed above, GLS knock out in cones likely impacts their function, in accordance with the data presented at ARVO 2024.

Recommendations for improving the writing and presentation.(1) In the Discussion, lines 458-466, it's incorrect to compare the importance of glucose metabolism to GLS-dependent pathway to photoreceptors in this way. An alternative explanation: glucose metabolism is so important that the system has many redundancies, e.g. HK1 exists in addition to HK2, thus single gene KO leads to no phenotype. The only fair comparison is nutrient deprivation, e.g. taking out glucose or glutamine from retina explants (Punzo et al., 2009).

The reviewer makes an excellent point. While we do not see an upregulation of GLS2 in the retina or rod PRs upon GLS knockout (Figure 1-figure supplement 1 D and E), loss of *Gls* in rod PRs does alter the expression of many metabolism-related genes (Figure 4-figure supplement 1). We alluded to these data and the reviewer’s point in the second paragraph of the discussion: “In any of these transgenic mouse models, PRs may use other transporters to take up fatty acids or glucose or rewire their metabolism to maintain metabolic homeostasis and stave off degeneration (Subramanya et al., 2023; Wubben et al., 2017). Our data show that any metabolic reprogramming that is occurring in the cKO mouse retina appears unable to significantly circumvent the significant and rapid PR degeneration suggesting the importance of Gln catabolism in rod PRs. Furthermore, inducing GLS knockdown in mature PRs also demonstrated rapid PR degeneration (Figure 3).”

In the revised article, we have amended these sentences to include the importance of metabolic redundancies. “In any of these transgenic mouse models, PRs may use other transporters to take up fatty acids or glucose, rewire their metabolism, or utilize metabolic redundancies to maintain metabolic homeostasis and stave off degeneration (Subramanya et al., 2023; Wubben et al., 2017). Our data show that any metabolic reprogramming that is occurring in the cKO mouse retina appears unable to significantly circumvent the significant and rapid PR degeneration suggesting the importance of Gln catabolism in rod PRs. Furthermore, inducing GLS knockdown in mature PRs also demonstrated rapid PR degeneration (Figure 3).”

(2) Please discuss the mosaic activity of Rho-cre used in this study, as described in the original study (Le et al 2006). Line 221 (Li et al 2005) seems to be a different *Rho-Cre* created by a different group. Please make sure the citation is correct and consistent.

We apologize for the confusion and have corrected the reference on line 221 to Le et al, 2006. The reviewer is correct that the original report (Le at al. 2006) demonstrated a mosaic of Cre-mediated recombination in rod photoreceptors and rod bipolar cells in the mouse line that had the shorter (0.2 kb) mouse opsin promoter-controlled Cre. In contrast, this same report showed only Cre-mediated recombination in rod photoreceptors in another line that utilized a long (4.1 kb) mouse opsin promoter-controlled Cre. We have published using this latter promoter-controlled Cre recombinase in at least 5 different mouse models (Wubben et al. 2017; Weh et al. 2020; Weh et al. 2023; Subramanya et al. 2023; the current report), and in all these models, we observe clear and consistent knockout by immunofluorescence only in rod photoreceptors with residual protein in cones and no significant change in protein expression in the INL where bipolar cells reside. Western blots confirm the reduction in protein expression.

(3) The authors should provide representative images of retina cross-sections for key rescue data (Figure 6G&H).

As requested by Reviewer 3, representative histology images of retina cross-sections for the ISRIB and Asn rescue experiments in *Gls* cKO mice at P21 are now included in the manuscript in Figure 6 – figure supplement 3.

Minor corrections to the text and figures.(1) Spell out Gln in the Abstract when used for the first time.

We have included glutamine (Gln) in the abstract upon first use.

(2) Line 433, Figure 6G should be 6H.

Thank you for the correction, the manuscript has been updated.